

# Radio frequency interference mitigating hyperspectral L-band radiometer

Peter Toose[1], Alexandre Roy[2], Frederick Solheim[3], Chris Derksen[1], Tom Watts[4], Alain Royer[2] and Anne Walker[1]

[1] Climate Research Division, Environment and Climate Change Canada, Toronto, Ontario, M3H 5T4 Canada.
[2] Centre d'Applications et de Recherches en Télédétection, Université de Sherbrooke, Sherbrooke, Quebec, J1K 2R1, Canada
[3] Dakota Ridge Research and Development, Boulder, Colorado, 80303, United States.
[4] Department of Geography, Northumbria University, Newcastle upon Tyne, NE1 8ST, United Kingdom.

*Correspondence to*: Peter Toose (Peter.Toose@canada.ca)

**Abstract.** Radio Frequency Interference (RFI) can significantly contaminate the measured radiometric signal of current spaceborne L-band passive microwave radiometers. These spaceborne radiometers operate within the protected passive remote sensing and radio astronomy frequency allocation of 1400-1427 MHz, but despite this are still subjected to frequent RFI intrusions. We present a unique surface-based/airborne hyperspectral 385 channel, dual polarization, L-band Fourier transform, RFI detecting radiometer designed with a frequency range from 1400 through ≈1550 MHz. The extended frequency range was intended to increase the likelihood of detecting adjacent RFI-free channels to increase the signal, and therefore increase the thermal resolution, of the radiometer instrument. The external instrument calibration uses three targets (sky, ambient, and warm) and validation from independent stability measurements shows a mean absolute error (MAE) of 1.0 K for ambient and warm targets, while the MAE is 1.5 K for sky. A simple but effective RFI removal method which exploits the large number of frequency channels is also described. This method separates the desired thermal emission from RFI intrusions, and was evaluated with synthetic microwave spectra generated using a Monte Carlo approach and validated with surface-based and airborne experimental measurements.

**Keywords:** surface-based / airborne radiometer, L-Band, radio frequency interference, RFI mitigation

## 1 Introduction

A number of spaceborne L-Band passive microwave radiometer missions were successfully launched in recent years for global monitoring of soil moisture and sea surface salinity. The European Space Agency Soil moisture and Ocean Salinity (SMOS) mission (Kerr et al., 2010) was launched in November 2009 and continues to operate. The NASA-SAC/D Aquarius mission acquired L-Band observations between September 2012 to July 2015 (Lagerloef et al., 2013), and the NASA Soil Moisture Active Passive (SMAP) satellite was launched in January 2015 (Entekhabi et al., 2015). In addition to soil moisture objectives, these missions also provide useful measurements for cryospheric applications including monitoring the



freeze/thaw state of the land surface (Rautiainen et al., 2016; Rautiainen et al., 2014; Roy et al., 2015), estimating snow density and ground permittivity (Schwank et al., 2015; Lemmetyinen et al., 2016), and retrieving the thickness of thin sea ice (Kaleschke et al., 2016; Kaleschke et al., 2012).

Even though the SMOS, Aquarius, and SMAP radiometer bandwidths fall within a protected band (1400-1427 MHz), significant levels of radio frequency interference (RFI) caused by anthropogenic sources of radiation are commonly observed in satellite L-Band measurements (Oliva et al., 2016; Le Vine et al. 2014;Piepmeier et al., 2014; Askoy and Johnson, 2013). The United Nations provision 5.340 of Radio Regulations of the International Telecommunication Union - Radiocommunications Sector (ITU-R), has regulated that the frequency allocation of 1400-1427 MHz be dedicated to

passive remote sensing from space and radio astronomy research and that all other emissions within this band be prohibited (ITU, 2012). Illegal sources, in-addition to spurious and harmonic emissions and other unwarranted transmissions, often violate these reserved bands producing RFI. The natural thermal emissions in these protected wavebands are orders of magnitude lower in power than active RFI sources; therefore, such RFI intrusions can contaminate and even blind the passive observations. While a number of different hardware and processing approaches have been explored and implemented

(Guner et al., 2007; Anterrieu, 2011; Forte et al., 2011; Pardé et al., 2011; Misra et al., 2009), it is a continuing challenge to identify, and if possible, detect, avoid or mitigate RFI from L-band radiometer observations (satellite and airborne).

Several surface-based and airborne L-band radiometers have been in operation for calibration/validation algorithm development activities during the pre- and post-launch periods for SMOS, Aquarius and SMAP (Misra et al., 2013; Pardé et

al., 2011; Delwart et al., 2008; Rautiainen et al., 2008). With this in mind, a hyperspectral L-band radiometer system was developed by Radiometrics Corporation©, Boulder, Co., and delivered to Environment and Climate Change Canada's (ECCC) Climate Research Division in 2010, that has the capability to detect broadband RFI and mitigate narrowband RFI over an expanded bandwidth with high frequency resolution, in a package that can be easily mounted on both tower and airborne platforms. This advanced dual polarization, hyperspectral L-band radiometer can be operated in the 385 channel

hyperspectral mode which spans 1400 through ≈1550 MHz. The objectives of this paper are to provide an overview of the radiometer, characterize the performance and calibration, and describe a simple but effective method of separating out the thermal spectrum channels from those contaminated by narrowband RFI. Two evaluation datasets were utilized: ground-based measurements made through the 2014-2015 winter season in Saskatchewan, Canada to monitor the freeze/thaw state of the surface soil layer, and airborne measurements over sea ice acquired in the Canadian Arctic in April 2011.




## 2 Radiometer system

### 2.1 Radiometer architecture

During the initial design stage of the hyperspectral radiometer, it was known that RFI was commonly present in the protected band 1400-1427 MHz. Therefore, the radiometer was designed with a bandwidth expanded outside the protected band with

the intention of improving the ability to detect adjacent RFI-free channels. The full radiometer bandwidth observes the 1375 to 1575 MHz spectrum in 512 channels (≈391 kHz width) at both horizontal and vertical polarizations. The system described herein consists of a downconverter module, a digital processor module, and a conformal antenna with interconnect and power cables.

The radiometer down-converter/receiver serially measures both antenna polarizations, a 50 ohm ambient load, and a calibrated noise diode that can be modulated with a cpu-controlled programmable attenuator. A bandpass filter follows the first antenna isolator to reduce intermodulation that might occur at the input of the first Low Noise Amplifier (LNA). A synthesized local oscillator and selectable bandpass filters allow either a 25 MHz bandpass to be located within the 1375 to 1575 MHz receiver band, or for the full 200 MHz to be sampled. The down-converted power is split into two outputs with

the first being a detected video output for operation as a Dicke radiometer. The second output is an Intermediate Frequency (IF) power output that is fed to a digital signal processor via a 400 megasamples/sec 14 bit analog-to-digital (A2D) sampler for subsequent Cooley-Tukey type Fourier transformation. The Fourier transformation is accomplished in a Xilinx Field Programmable Gate Array (FPGA), which includes a blanking algorithm followed by the transform. Ten microsecond 1024 (512 horizontal and 512 vertical channel) wide spectra are accumulated and passed over a PCI bus to a single board

computer. The computer also controls the functions of the receiver.

The internal components of the receiver are mounted to a thick aluminum plate for stability and thermal uniformity. The plate is temperature controlled by Peltier junctions to within ±0.03°C of 35°C, and is in an enclosure separate from the digital signal processor. The system has a dynamic range of 70 dB. Because of filter roll-off, windowing and Nyquist effects,

the full 512 channels, 200 MHz receiver bandwidth, for each polarization output from the Fourier Transform are not suitable for analysis, instead only 385 channels and ≈150 MHz bandwidth (1400 to 1550.5 MHz) are utilized. The 385 calibrated brightness temperature channels are each ≈391 kHz in width. The narrow bandwidth of the channels greatly enhances the ability to detect narrowband RFI intrusions compared to what would be realized if an intrusion fell in a broadband (e.g. 25 MHz) channel. The L-band radiometer system component and measurement specifications are listed in Table 1.


The antenna was designed to be compact (51 x 51 x 6.5 cm and ≈5 kg), for easy mounting on ground and airborne platforms. A shallow antenna depth was necessary for aircraft mounting to reduce drag, therefore, a 19 element air loaded conformal muffin tin design was utilized (Fig. 1). The loss of this antenna and its combiners is less than 0.5 dB. The temperature of the



antenna and of the two antenna-to-down converter cables are measured and recorded for system temperature and gain corrections. The antenna was designed with a 30º half-power beamwidth (-3 dB), with side lobes below –20 dB. The angle of incidence can be manually adjusted between measurements when in operation on the ground, and is fixed at 40 degrees when operated on an aircraft (SMAP viewing mode). At 40 degrees, the large beamwidth of the radiometer antenna produces a

footprint depth that approximates the height of the radiometer antenna above the surface, and a width that is approximately ¾ the size of the depth.

The radiometer has 6 video states, which include observing the vertical antenna, vertical antenna plus noise diode, horizontal antenna, horizontal antenna plus noise diode, load and load plus noise diode in one integration cycle. Each sample takes 0.5-

milliseconds, and each time a state changes the radiometer waits for 4 samples and then integrates for 22 samples. The radiometer observes all six states 50 times, with a ≈60-millisecond delay between each integration cycle to record system information, producing one scene measurement every ≈3.9 seconds for all channels and both polarizations. There can be some variance in the integration time due to the overhead associated with the thermal control and the settings of the synthesizer (0 – 1.5 seconds).

## 2.2 Radiometer calibration

A calibrated brightness temperature ($T_B$) is computed for each channel and polarization. The radiometer determines the $T_B$ of the field of view (FOV) by measuring the receiver video voltage difference between the FOV ($V_{sky}$), and the internal load observations ($V_{load}$), and comparing this with the voltage difference realized by turning on the noise diode while at the

internal load ($V_{loadND}$). This noise diode voltage enhancement is the measure of gain that determines the temperature difference between the internal load and $V_{sky}$.

$$V_{load} = g(T_{RCV} + T_{load} + Offset)^{\alpha} \tag{1}$$
$$V_{loadND} = g(T_{RCV} + T_{load} + Offset + T_{ND})^{\alpha} \tag{2}$$
$$V_{sky} = g(T_{RCV} + T_{sky})^{\alpha} \tag{3}$$

The noise diode values ($T_{ND}$), are the gain reference for the radiometer, and scale the voltage differences between the internal load measurements and the observations out the antenna. The offset values ($Offset$), calibrate the internal load temperature as seen by the radiometer receiver. $T_{ND}$ and $Offset$ are calculated using the following fixed calibration

parameters: ($\alpha$, $T_{ND}(0°C)$, $T_{ND}TC$, $Offset(0°C)$ and $OffsetTC$). However, due to the temperature dependence of gain and offset, these fixed values must first be temperature corrected using the case temperature in which the receiver is housed,


($T_{case}$ in °C), which fluctuates with the outside ambient operating temperature surrounding the receiver/antenna and all cables:

$$T_{ND} = T_{ND}(0°C) + T_{ND}TC \cdot T_{case} \qquad (4)$$

$$Offset = Offset(0°C) - OffsetTC \cdot T_{case} \qquad (5)$$

To determine the gain and offset values, measurements of at least two targets of known black body temperatures are required. The following equations are used to derive $T_B$ from these measured data for each channel and each polarization (512 vertical channels (V-pol), 512 horizontal channels (H-pol)), from the observed scene. These equations take into account the internal temperatures of the receiver and the internal load which are both set to 35°C:

To calculate $T_B$, the gain (g) is calculated,

$$g = \left[ \frac{V_{loadND}^{\frac{1}{\alpha}} - V_{load}^{\frac{1}{\alpha}}}{T_{ND}} \right]^{\alpha} \qquad (6)$$

The receiver temperature is then calculated,

$$T_{RCV} = \left( \frac{V_{load}}{g} \right)^{1/\alpha} + T_{load} - Offset \qquad (7)$$

Then $T_B$ is,

$$T_B = \left( \frac{V_{sky}}{g} \right)^{1/\alpha} + T_{RCV} \qquad (8)$$

The field calibration measurement techniques and post-processing procedures will be explained further in Sect 3.2.

## 3 Methods

### 3.1 Sites and data collection

The L-Band radiometer calibration accuracy and RFI mitigation approach was evaluated using ground-based measurements of brightness temperature over varying soil moisture and freeze/thaw conditions throughout the 2014-2015 winter season as part of an ECCC led L-Band freeze/thaw detection campaign. Continuous radiometer measurements were observed from


≈2.75 m above the ground mounted on a moveable platform, at the University of Saskatchewan's Kernen Crop Research Farm within the city of Saskatoon (52.149° N; 106.545° W), (Roy et al., submitted). These temporally continuous measurements were augmented with monthly visits, and distributed radiometer measurements, across the Kenaston/Brightwater Creek soil monitoring network in a rural area south of Saskatoon (≈51.3° N; ≈106.5° W). Between

October 21$^{st}$, 2014 and April 14$^{th}$, 2015, 16 three-target calibrations (sky, ambient black body, and heated black body: see next section for descriptions of targets) were conducted. During the campaign, regular radiometer stability-check measurements of each calibration target: sky (91 measurements), ambient (19 measurements) and heated (13 measurements) were recorded. The calibration measurements were used to calibrate the radiometer, while the stability-check measurements were used to check that the instrument was stable and that the current calibrations were still valid.

The L-band radiometer was also deployed in the Canadian Arctic during the Polar Airborne Measurements and Arctic Regional Climate Model Simulation Project (PAMARCMIP) in 2011. During the campaign, the L-band radiometer was mounted aft-viewing (along-track), on the Alfred Wegener Institute's Polar 5 research aircraft, with a 40º incidence angle. During one of the research flights, the aircraft flew at an altitude of ≈100 meters over first year sea ice in Nares Strait

(78.797° N; 74.059° W), which would normally produce an approximate radiometer footprint of ≈96 m x ≈74 m if the radiometer antenna were stationary. However, the aircraft platform was travelling at an average ground speed of ≈65 meters per second, and with the radiometer's approximate 4 second integration cycle, the average footprint size was ≈74 m wide (across-track), and ≈356 m in the along-track direction of travel.

Both sets of measurements were utilized to assess the effectiveness of the RFI removal algorithm. Both constant and intermittent continuous wave narrowband RFI was observed during all flights and calibrations during the PAMARCMIP 2011 campaign. Some similarities in detected patterns of RFI were noted between the PAMARCMIP 2011 campaign and other research campaigns. It was suspected that at least some of the RFI detected by the hyperspectral L-Band radiometer emanates from the radiometer system itself. To investigate, the radiometer system was placed in a closed metal shipping

container, acting like a Faraday cage, and different components of the system (generator, power supply, digital signal processing computer) were removed from the container one-by-one to investigate the change in measured TB. It was noted that both the digital signal processing computer and the radiometer receiver were sources of intermittent narrowband L-Band RFI. Mitigation of these RFI sources involved housing the computer in an enclosed metal box, while the position of the radiometer receiver was mounted behind the antenna. These changes reduced the intensity and frequency of occurrence of

RFI detected, but it did not completely remove their intermittent contribution. However, due to the hyperspectral nature of this L-band radiometer, the intermittent narrowband RFI can be further mitigated through post-processing of the data.




### 3.2 Calibration measurement procedure/post-processing

The absolute accuracy of a radiometer is dependent on the internal calibration procedure, the uncertainty of the external calibration, and the inherent stability of the radiometer's internal electronics. An external calibration is necessary to account for losses and reflections in components of the radiometer system outside of the receiver and to account for changing

environmental variables. Surface-based and airborne radiometer systems are exposed to a stressful and ever-changing environment: the system is typically not mounted permanently, but is assembled and disassembled for each campaign; the power is turned on and off repeatedly each day and environmental temperature can vary significantly between measurements.

During aircraft campaigns there is a risk that strong RFI may be present at airports due to airport radars and air route surveillance (Hallikainen et al., 2010; Skou et al., 2010; Le Vine, 2002), which may pose a challenge when trying to calibrate at these locations. Therefore, a field transportable two temperature calibration target was built to couple with the ECCC radiometer antenna. The calibration system was built into a 70x70x70 cm shipping container that splits in-half with one side heated and the other side an ambient black body target. A fan circulates air through the open cell black body foam

in which 5 precision temperature sensors are embedded. The heated "warm" side is controlled at ≈343 K, while the "cold" side is controlled by the outside ambient temperature. A three-point calibration procedure using the sky, ambient, and heated warm targets was developed to account for the radiometer's calibration non-linearity and improve the accuracy over the full range of measured $T_B$. The reference sky $T_B$ at L-Band was considered to be ≈5 K for polarizations (Pellarin et al., 2016; Lemmetyinen et al., 2016).

We utilize the ground-based measurements from Saskatoon to illustrate the calibration procedure. The post-processing of the calibration measurements involves the optimization of calibration coefficients, which takes place in two steps: 1) First, all five calibration coefficients ($\alpha$, $T_{ND}(0°C)$, $T_{ND}TC$, $Offset(0°C)$ and $OffsetTC$), are optimized for all 385 channels and both polarizations, using all 16 three-target calibration measurements acquired between October 2014 and April 2015. This

first step was done to take into account the changes in the environmental operating temperature ($T_{case}$), of the radiometer between calibration dates. Measured $T_{case}$ from all 16 three-target calibrations ranged from -18.1°C to 23.5°C for the entire campaign. To assess the sensitivity of the radiometer to variations in $T_{case}$, calibrated $T_B$s were calculated for the November 9[th] ambient calibration measurements, computed using the exact same set of coefficients, with the exception of the $T_{case}$ value. The minimum and maximum observed $T_{case}$ values were used as inputs (-18.1 and 23.5 °C), leading to a difference of

≈10 K (269.4 K versus 259.3 K), highlighting the importance of applying a temperature correction. After optimizing the coefficients with $T_{case}$, a robust local regression using weighted linear least squares and a 2[nd] degree polynomial model was applied to create a smooth calibration curve for each coefficient for all channels. Applying the 2[nd] degree polynomial model to the calibration coefficients removes some of the inter-channel variability, however, it can also be used to apply





meaningful coefficients to RFI affected channels. If a narrow group of channels is contaminated with intermittent RFI during the calibration procedure (Fig. 2), the smoothed polynomial calibration curve can apply adjacent calibration coefficients to the RFI affected channels so that usable data can still be acquired at these frequencies.

The results from completing step 1 of the post-processing are used to fix the coefficients for $OffsetTC$, $T_{ND}TC$ and α as constants. 2) Secondly, the next stage of the post-processing involves using the optimization results for the coefficients $T_{ND}(0°C)$ and $Offset(0°C)$ from the first post-processing step as a first guess, to run the inversion model again, this time to produce a set of new calibration coefficients for $T_{ND}(0°C)$ and $Offset(0°C)$, for each date of the 16 three-target calibrations to correct for inherent instability of the radiometer's internal electronic components over the course of the campaign. The

optimized values for $T_{ND}(0°C)$ and $Offset(0°C)$, are then interpolated in time between calibration measurements with simple linear regression. These interpolated calibration coefficients are then used to calculate the final science ready $T_B$ for all measurements recorded between the calibration dates, and were also applied to all radiometer calibration stability-check measurements to assess the calibration accuracy for the campaign.

**3.3 Hyperspectral RFI mitigation approach**

A number of L-Band RFI identification and mitigation methods have previously been proposed and developed into hardware for both airborne and satellite systems (Guner et al., 2007; Anterrieu, 2011; Forte et al., 2011; Pardé et al., 2011; Misra et al., 2009), but the hyperspectral system described in this paper has the capability of identifying RFI-free channels over an expanded 1400-1550 MHz spectrum. Assuming that the natural $T_B$ of a scene is a Gaussian distribution across the

frequency range 1400-1550 MHz, a simple but effective method of separating out the thermal spectrum from RFI-contaminated channels is to sort the $T_B$s in ascending order. The thermal channels sort to the low values and RFI-contaminated channels are then identified where the brightness begins to rise out of the expected thermal spectrum. To find the mean $T_B$ of this spectrum a 3$^{rd}$ order polynomial of sort-rank versus $T_B$ is calculated, and the 2$^{nd}$ derivative of the slope of this cubic polynomial is derived. For a clean thermal spectrum with random noise, the mean value is a close approximation

of the $T_B$ at the inflection point where the 2$^{nd}$ derivative goes from negative to positive. RFI is assumed to reside in those channels with $T_B$ higher than this value. Fig. 3 illustrates the horizontal polarized $T_B$s of one integration cycle of the ambient calibration target (251 K) shown in Fig. 2, with the data sorted into ascending order represented by the blue-dashed line. The red dot in Fig. 3 is the inflection point representative of the mean $T_B$ of the Gaussian distribution. To the right of the inflection point are the natural thermal spectrum values larger than the mean $T_B$ value (the high side of the Gaussian

distribution of thermal energies), and the RFI affected channels. To the left are the values of the natural thermal spectrum smaller than the mean value. The RFI mitigation approach does not work when the spectrum being analyzed is not part of a normal distribution. This either results from extreme RFI contamination, or from limiting the application of the RFI mitigation approach to a narrow bandwidth (such as 1400-1427 MHz), with too few channels creating a uniform, short-tailed



distribution when the scene is RFI-free. This RFI mitigation approach is applied to each polarization separately, typically for all 385 channels in the 1400-1550 MHz spectrum. All subsequent temporal averaging of the $T_B$ measurements is then applied to the RFI mitigated data.

### 3.4 Sensitivity analysis of the RFI mitigation approach using modelled data

To evaluate and validate the RFI mitigation approach, a Monte Carlo experiment was conducted to create synthetic L-Band spectra with random RFI peaks. The experiment tested two main characteristics of RFI that can affect the performance of the RFI mitigation algorithm: 1) the number of peaks within the spectrum (#peaks), and 2) the bandwidth of those peaks ($\omega$), represented by the number of channels (1 $\omega$ = 390.625 kHz). For each different combination of #peaks (0-20), and $\omega$ (1-3-5-10 channels), 1000 replicate simulations were run and the RFI removal algorithm was applied to the synthetic spectra. The

mean $T_B$ of the synthetic scene was set to 250 K with a random noise of 3.6 K added to all channels, representing the typical standard deviation measured by the radiometer from an unfiltered single integration cycle measured across the 1400-1550 MHz spectrum. The peak amplitudes were randomly set based on absolute values of a normal distribution with mean 0 and standard deviation 100. For all 1000 replicates of the different combinations of #peaks and $\omega$, a mean $T_B$ value ($T_{B-mean}$) of the RFI mitigation results and a standard deviation were calculated. The performance of the RFI mitigation approach was

assessed by comparing the difference between the mean RFI mitigation results to the 250 K mean $T_B$ of the synthetic spectrum. If the RFI mitigation results were within 2 K of the set 250 K mean of the synthetic spectrum, then the RFI mitigation approach was considered successful. A 2 K threshold was chosen because it is slightly larger than the combination of the calibration accuracy ($\approx$1.5 K; see section 4.1), and the radiometer's radiometric resolution for the entire 150 MHz spectrum (0.13 K; see Table 1), added together.

### 4 Results

In this section, we present the results of the calibration accuracy assessment and radiometer stability observed during the 2014-2015 Saskatchewan soil freeze/thaw detection campaign. A sensitivity analysis using modeled data to assess the performance of the hyperspectral RFI mitigtation technique under varying degrees of RFI saturation  using a Monte Carlo

approach is provided. The sensitivity analysis is followed by two examples of the hyperspectral RFI mitigation technique applied to L-Band surface-based radiometer measurements of unfrozen bare soil in Saskatchewan, and of first year sea ice from airborne observations in the Canadian Arctic.



### 4.1 Radiometer calibration results and instrument stability evaluation

The calibration accuracy and radiometer instrument stability were examined using the surface-based radiometer measurements recorded in Saskatchewan from October 2014 to April 2015. Figure 4 shows the optimization results of the first step of the calibration post-processing, where all calibration coefficients ($T_{ND}(0°C), Offset(0°C), T_{ND}TC, OffsetTC$

and $\alpha$) are optimized for $T_{case}$ at the same time using all 16, three-target calibrations. The variability in the optimized coefficient results (dashed-green line in Fig. 4), is likely a direct result of the varying amounts of RFI present during each calibration date, emitted by both external sources and internal system components. This RFI induced noise in the optimization results is smoothed using a robust local regression using weighted linear least squares and a 2$^{nd}$ degree polynomial model (solid blue line in Fig. 4).

Using the results of the smoothed (blue line in Fig. 4) calibration coefficients, $\alpha$ , $T_{ND}TC$ and $OffsetTC$ are held constant, while the smoothed results for the coefficients $T_{ND}(0°C)$ and $Offset(0°C)$ are used as a first guess, to run the inversion model again, this time to produce a set of new calibration coefficients for $T_{ND}(0°C)$ and $Offset(0°C)$, for each date of the 16 three-target calibrations to correct for sensor drift of the radiometer over course of the campaign (Fig. 5: red dots - for the

1451 MHz channel). The $T_{ND}(0°C)$ optimization results in Fig. 5 identify some random sensor drift at the beginning of the season with a large step change shift observed between the January 14$^{th}$ and January 31$^{st}$ calibrations. This step change was possibly related to a power supply failure that occurred on January 18$^{th}$, (replaced on January 21$^{st}$). The $T_{ND}(0°C)$ calibration coefficients then trended upward, stabilizing near the end of March. The $Offset(0°C)$ optimization results in Fig. 5 identify a more random pattern throughout the campaign, with the exception of a noticeable drop in $Offset(0°C)$ value in February,

which remains unexplained.

To account for radiometer drift between external calibrations, $T_{ND}(0°C)$ and $Offset(0°C)$ were interpolated in time with simple linear regression, and these interpolated coefficients were then used to calculate daily calibrated $T_B$ (Fig. 5: black lines). In most instances, the interpolated coefficients are representative of the gradual radiometer sensor drift due to the

inherent instabilities of the internal electronics. However, there are exceptions, such as when the radiometer sensor under-goes a step change in stability, as took place between January 14$^{th}$ and 31$^{st}$, 2015. The change in calculated coefficients between the 14$^{th}$ and 31$^{st}$ calibration dates was several times larger than the difference in coefficients between all other dates, and thus the application of a simple linear regression to interpolate the coefficients in time between these dates may not be the best course of action because applying the interpolated coefficients implies a continuous drift in coefficients, instead of

the observed step-wise change in system stability. Therefore, all data that fell between these two calibration dates made use of the non-interpolated coefficients from the calibration date that was closest. This time series of calibration coefficients highlight the importance of regularly verifying the calibration of an instrument during a research campaign.



To evaluate the radiometer's calibration accuracy, the interpolated calibration coefficients were used to produce calibrated $T_B$ for measurements acquired during all radiometer stability-checks of the sky ($T_{Bsky}$), ambient black body targets and heated warm black body targets made at the Kernen Crop Research Farm and the Kenaston/Brightwater Creek soil monitoring network in Saskatchewan between October 2014 and April 2015. These stability-check measurements were

independent of those used to produce the calibration coefficients. The difference between the black body physical temperature and radiometer $T_B$ are calculated ($\delta T_{B\text{-}BB}$). Mean absolute error (MAE) of 1 K (Fig. 6 left) with ambient and warm targets and 1.5 K (Fig. 6 right) with sky measurements from both the Kernen and Kenaston sites ($T_{Bsky} = 5$ K from Lemmetyinen et al., 2016). The biases are lower than 0.3 K. For black body targets, only one point gives errors higher than 3K. This single occurrence is likely associated with an error in recording the black body physical temperature during the

stability-check. The higher error in the sky measurements could be related to the fact that the sky emission might vary slightly depending upon the observed portion of the sky (Pellarin et al., 2016). For example, the Milky Way can contribute galactic emission of up to 20 K. Further analysis of these effects will be evaluated in future campaigns. The response to soil freeze/thaw state and soil moisture fluctuations is on the order of 30 to 50 Kelvin, so a measurement time series with an MAE of less than 1.5K and minimal bias can be considered very robust for these applications.

### 4.2 Sensitivity analysis of the RFI mitigation approach using modelled data

The performance of the RFI mitigation approach was assessed by comparing the difference between the mean RFI mitigation results to the set 250 K mean $T_B$ of the synthetic spectra. If the RFI mitigation results were within 2 K, of the mean 250 K of the synthetic spectrum, then the RFI mitigation approach was considered successful. Figure 7 illustrates that for narrow RFI

peaks ($\omega = 1$), the RFI mitigation approach can retrieve the mean $T_B$ from the scene to within 2 K independent of the number of peaks (up to a maximum of 20 in this experiment). However, as both the bandwidth of the RFI peaks gets larger, and the number of peaks of larger bandwidth RFI sources increases, the performance of the RFI mitigation algorithm declines. Table 2 illustrates the results of the Monte Carlo experiment indicating the maximum bandwidth of the RFI affected channels (and proportion of total spectrum) that can affect the spectrum, while still being able to determine the mean $T_B$ value within 2 K

of the 250K synthetic spectra mean. The RFI mitigation approach was successful at retrieving an accurate mean $T_B$ with up to ≈13% of the spectrum contaminated with RFI.

### 4.3 Application of the RFI mitigation approach using experimental data

During the Saskatchewan L-Band soil campaign in 2014-2015, continuous surface-based measurements looking at a bare

soil surface, at 40° incidence angle were recorded. Steven's Hydra Probe soil moisture and temperature probes were installed within the FOV of the radiometer and were recorded using a CR1000 datalogger. The logger recorded data at 30 minute intervals, coinciding with intense repeated, short duration, narrowband RFI emission at ≈ 1450 MHz (V-pol = 1760 K; H-pol





= 5530 K) and ≈ 1550 (V-pol = 11 218 K; H-pol = 31 044 K), as illustrated in Fig. 8. These high $T_B$ peaks likely emitting from the logger show a significant contrast with normal continuous measurements where occasional relatively low RFI narrow peaks are seen (Fig. 9). This example allows for a comparison of the spectra over the same ground conditions with a significant RFI intrusion (Fig. 8), compared to one with nominal 'background' RFI present (Fig. 9).

For both cases, the mean broadband $T_B$ was calculated with and without the RFI mitigation algorithm with the results provided in Table 3. When the hyperspectral radiometer's RFI mitigation approach is applied to an observation affected by a powerful narrowband RFI source, it is able to return a mean $T_B$ value within <2 K of the same scene when only nominal background RFI is present. A broadband radiometer would have likely measured an average $T_B$ of 1761.3 K (H-pol) and

955.6 K (V-pol) while observing an RFI emission with this level of intensity over the same spectrum. This example with experimental data demonstrates the capability of the radiometer and RFI mitigation approach for determining a representative RFI-free $T_B$ value even in the presence of strong, narrowband RFI intrusions.

The RFI mitigation technique was also applied to airborne observations over first year sea ice, which contained both

narrowband (H-pol) and broadband (V-pol) RFI (Fig. 10). The difference between the measured $T_B$ with and without the application of the RFI mitigation approach is shown in Table 4 with a $T_B$ difference of 5.5 K at V-pol and 3.0 K at H-pol (1400-1550 MHz spectrum average $T_B$), highlighting that some RFI may add only a few Kelvin to the observed scene, and may be harder to detect utilizing a traditional broadband radiometer.

## 5 Conclusions

This paper describes a 385 channel hyperspectral L-band radiometer system. Using a three point calibration procedure, a mean absolute error (MAE) of 1 K for ambient and warm targets, and 1.5 K for a cold sky reference was determined. Because of the hyperspectral measurements, both narrowband and broadband RFI intrusions can be detected. A simple and straightforward RFI mitigation approach is effective in separating out powerful narrowband RFI-affected frequencies, to within 2 K of the natural thermal emission when the RFI is from narrowband sources contaminating up to ≈13% of the

observed spectrum. The performance of the RFI mitigation approach declines with the presence of increasing bandwidth (broadband) sources of RFI, however, the system still has the means to quantify the strength and type of interference with its compact design for mounting on both surface and airborne platforms, making this an ideal tool for calibration and validation activities of spaceborne L-band sensors. The examples of RFI intrusions during previous research campaigns also highlights the potentially high level of RFI exposure facing broadband L-Band radiometers in any environment, due to both powerful

illegal ground transmissions, and/or from subtle unintentional spurious emissions from complimentary research equipment.



## 6 Acknowledgments

The authors would like to recognize the important contributions of Mike Harwood, Ka Sung, and Mark Couture in their efforts towards the successful installation, programming, and processing of the airborne data. Thanks to Lauren Arnold, Arvids Silis and Bruce Cole for running the surface-based radiometer system for an entire winter season in Saskatchewan.
Thanks to staff at Radiometrics Inc. and Walter Strapp for all their work on the design, construction and testing of the hyperspectral L-Band radiometer. Thanks to the Canadian Space Agency and NSERC for financial support to the authors allowing for further development of the hyperspectral L-Band data processing procedures.

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


**Figures**

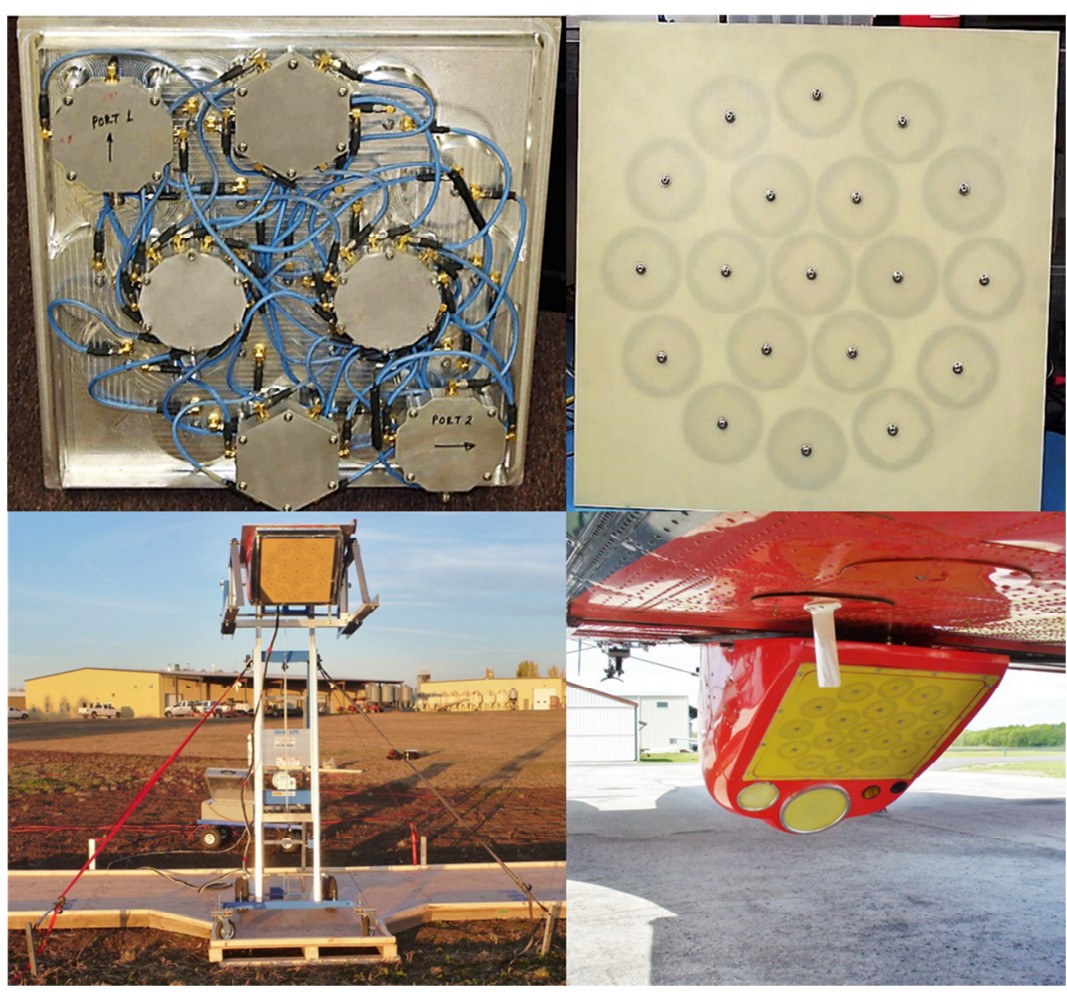

**Figure 1:** Conformal muffin tin antenna, back (top left) and front (top right), surface-based mobile platform (bottom left), airborne platform (bottom right; aircraft fuselage mount includes deployment with higher frequency radiometers).





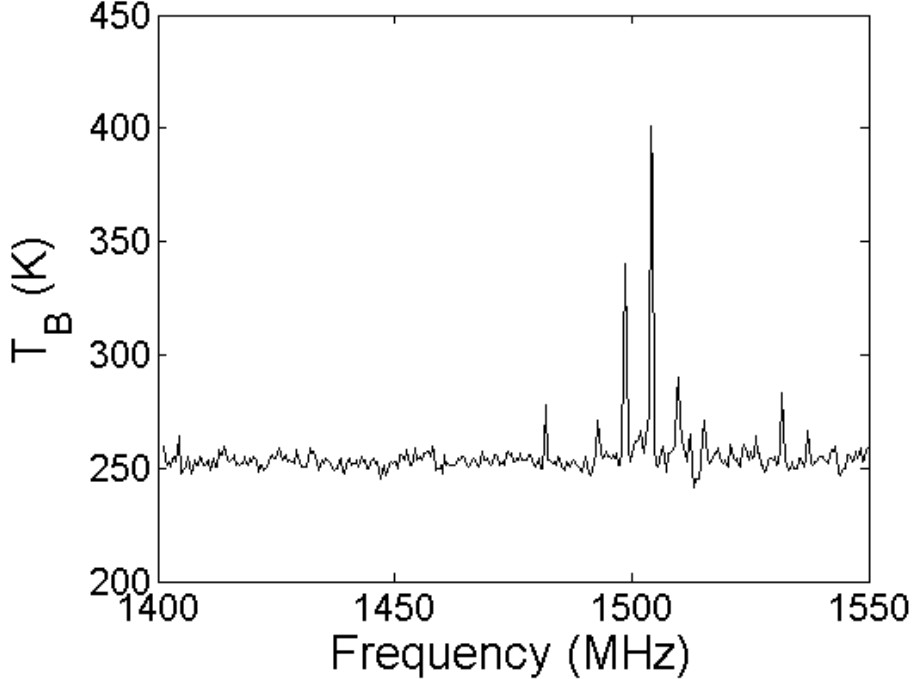

**Figure 2: Horizontally polarized $T_B$ spectrum of the ambient black body target (measured physical temperature of 251 K), during a stability-check measurement. Narrowband RFI peaks are observed around 1500 MHz. (11-02-2015 20:17 UTC).**





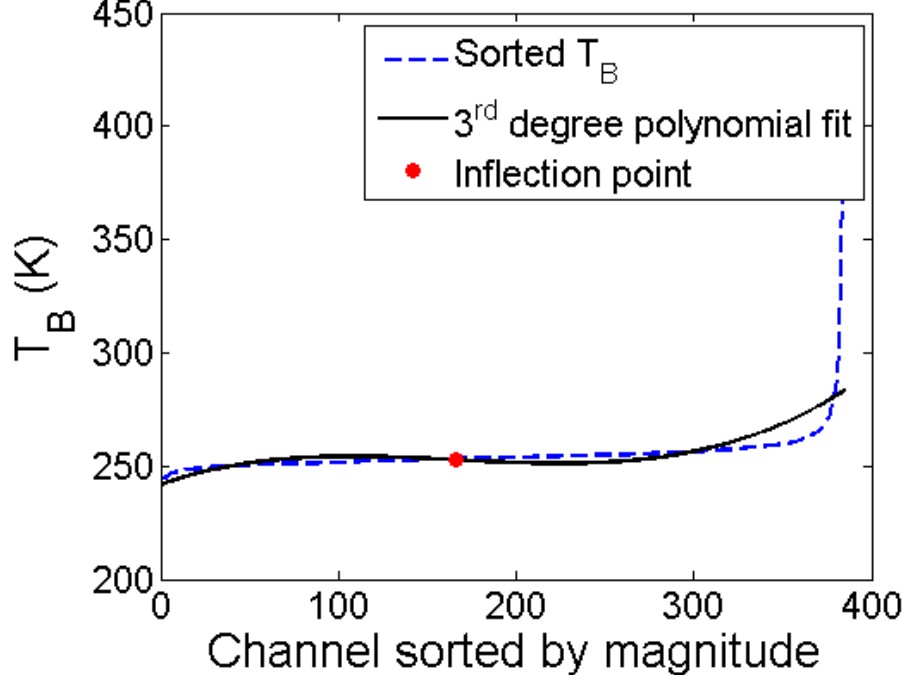

**Figure 3: Horizontally polarized $T_B$ of one integration cycle of the ambient calibration target (251 K; same spectrum as Fig. 2) sorted into ascending order. A 3rd order polynomial (black line) is used to approximate the curve of the sorted $T_B$ with the mean $T_B$ of the Gaussian distribution identified as occurring at the inflection point where the second derivative of this curve goes from negative to positive (red dot).**





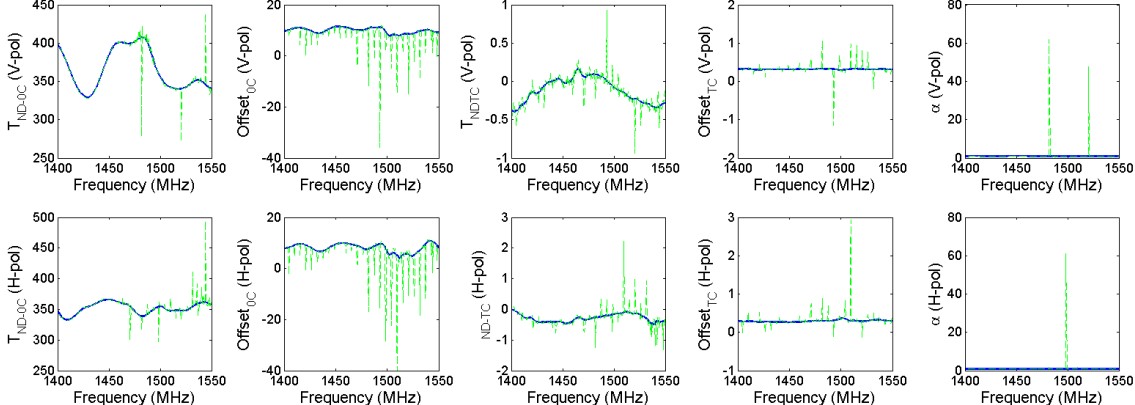

**Figure 4: Optimization results for the following calibration coefficients plotted from left to right: $T_{ND}(0°C)$, $Offset(0°C)$, $T_{ND}TC$,**
5   ***OffsetTC*** **and α for V-pol (top) and H-pol (bottom) for all frequencies. Green line: optimization results of all 16 calibrations; blue**
**line: robust local regression using weighted linear least squares and a 2nd degree polynomial model.**




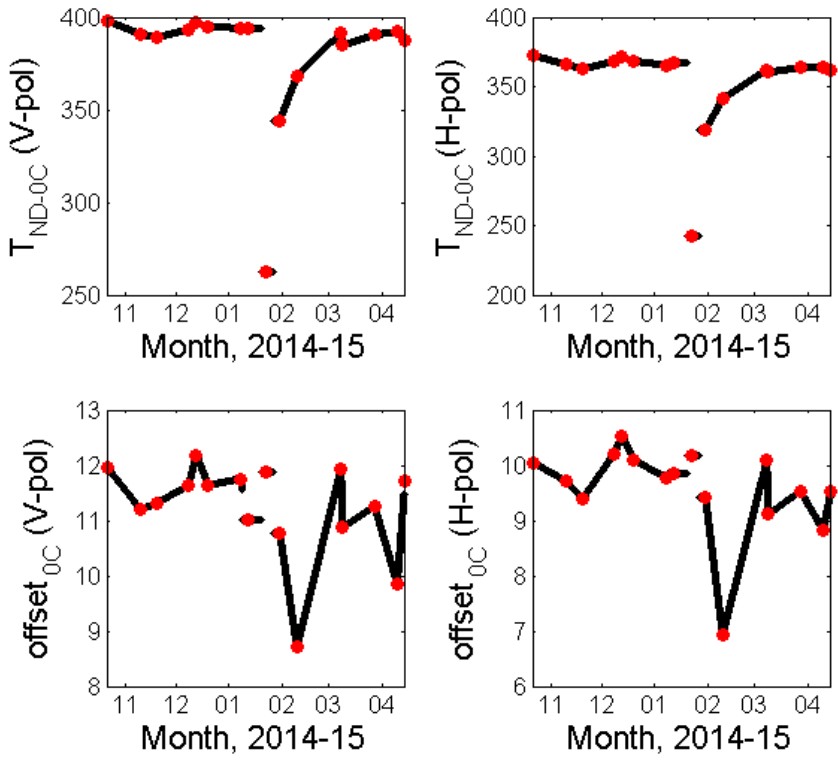

**Figure 5: Example of the V-pol (left) and H-pol (right) inverted calibration coefficient $T_{ND}(0°C)$ and $Offset(0°C)$ for the 1451 MHz channel during the 16 three-target calibration measurements (red dots) conducted between October 2014 to April 2015. The black line is the interpolation of calibration coefficient between measurement dates (red dots).**





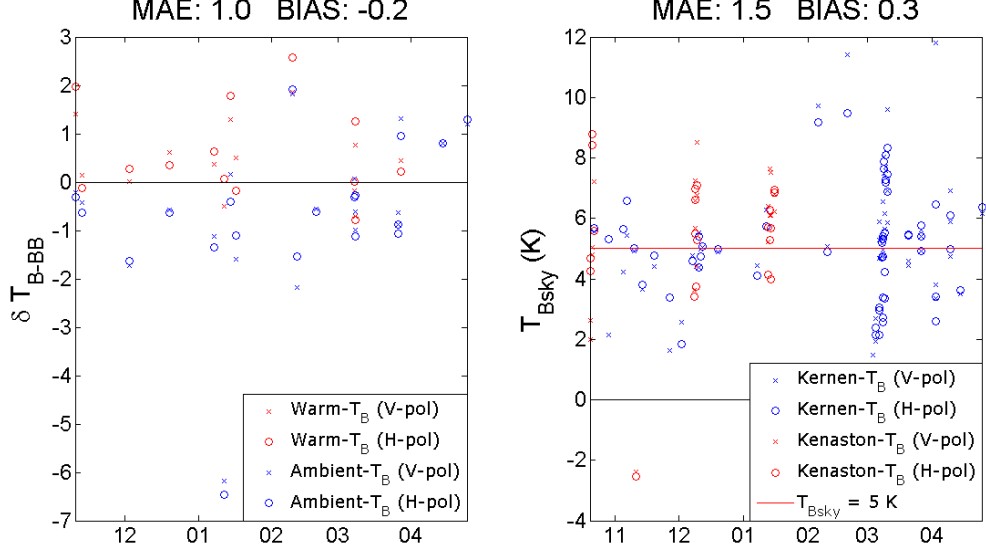

**Figure 6: The left plot illustrates the difference between measured black body physical temperature (BB) and the L-band radiometer measurement ($\delta T_B$). The right plot illustrates the difference between the theoretical $T_{Bsky}$ and the L-band radiometer measurements from the stability-check measurements. The x-axis is in time (month of the year).**



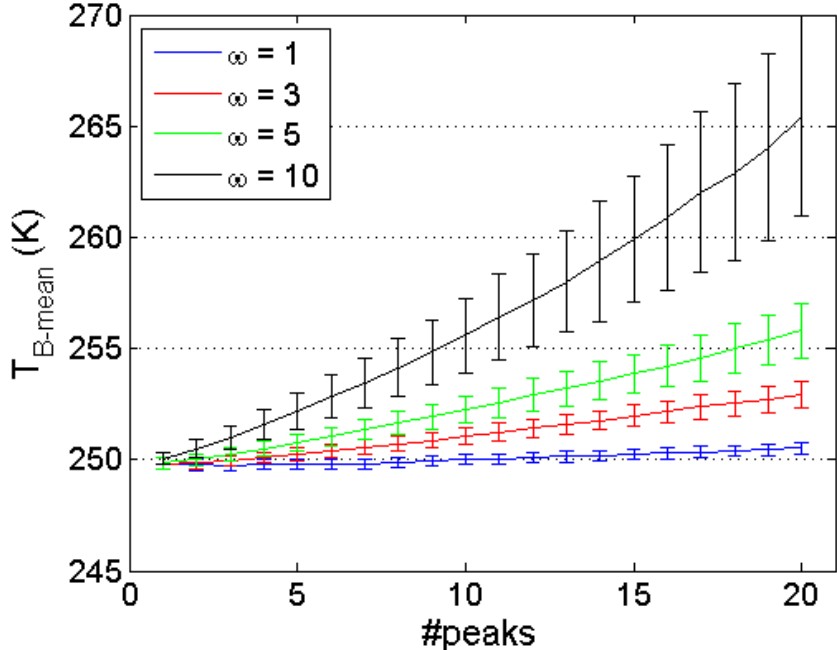

**Figure 7: For all 1000 replicates of the different combinations of peaks (0 up to 20) and bandwidths (1-3-5-10), a mean $T_B$ from the RFI mitigation results was calculated and plotted (solid lines) for comparison to the mean $T_B$ of the synthetic spectra (set to 250 K). The error bars show the standard deviation of the 1000 replicates for each scenario.**




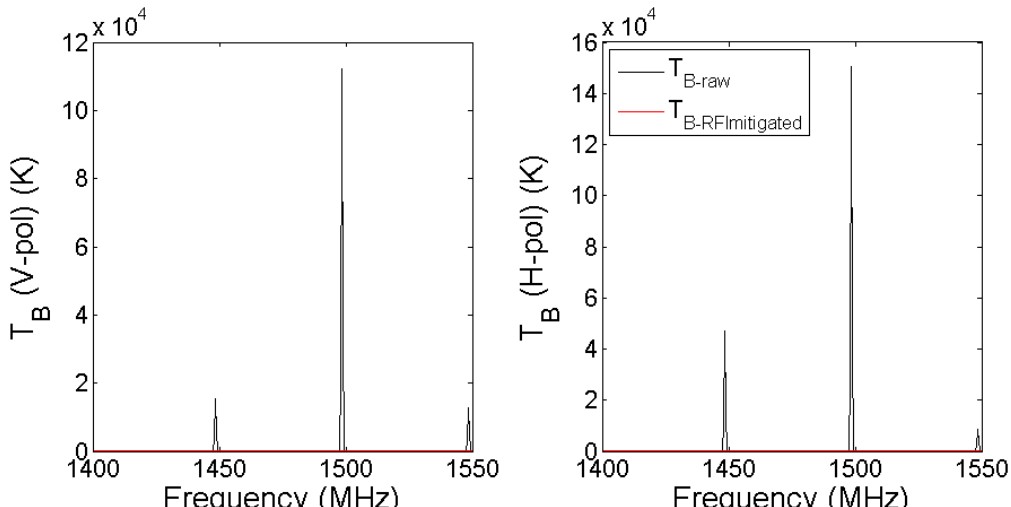

**Figure 8: V-pol (left) and H-pol (right) $T_B$ measured at Kernen crop research farm during datalogger RFI emission (25-10-2014 3:00:00 UTC).**




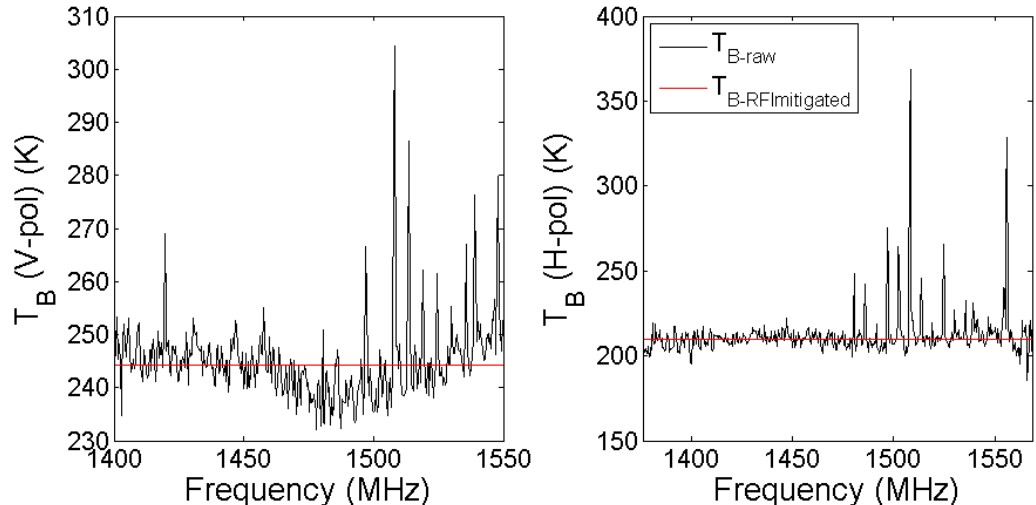

**Figure 9. V-pol (left) and H-pol (right) $T_B$ measured at Kernen crop research farm without datalogger RFI emission (25-10-2014 3:00:30 UTC).**





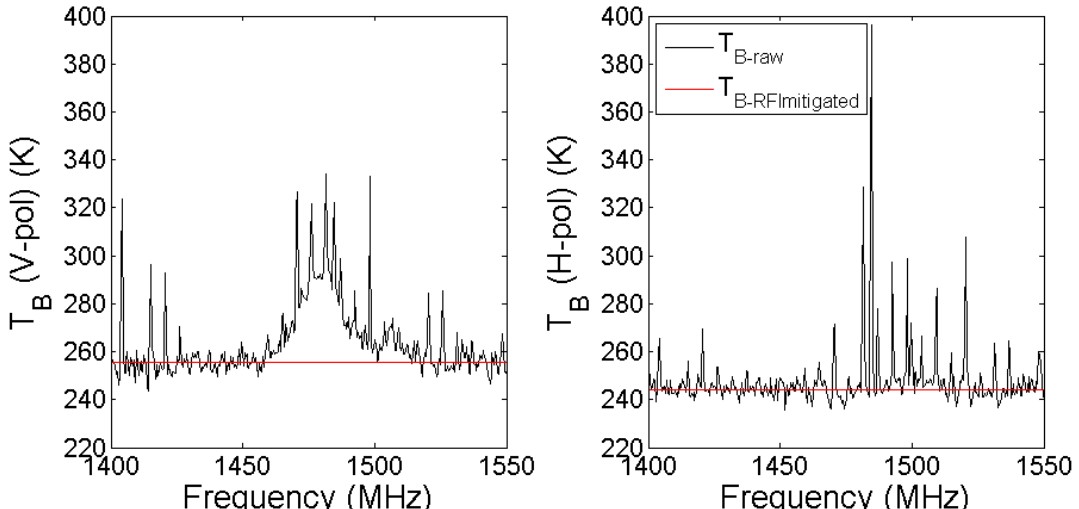

**Figure 10. V-pol (left) and H-pol (right) $T_B$ measured over first year sea ice in Nares Strait, Canadian Arctic (23-04-2011 16:25 UTC). Note the V-pol broadband RFI intrusion.**

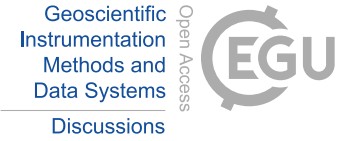

**Tables**

**Table 1: ECCC L-Band radiometer component and measurement specifications**

| Component | Specification |
|---|---|
| Radiometer receiver architecture | high sideband down conversion |
| Frequency range, GHz | 1.40 to 1.55 |
| Edge-to-edge IF bandpass, MHz | 150 MHz (hyperspectral mode) or 25 MHz (broadband mode); user selectable |
| Hyperspectral mode | 385 channels of ≈391 KHz width |
| Radiometric resolution, $\Delta T$, Kelvins | |
| single observation, single channel | 2.52 |
| single observation, 385 channels | 0.13 |
| Antenna HPBW, degrees | 30 |
| Sidelobes, -dB | -20 |
| Antenna type | conformal muffin tin antenna |
| Noise figure, dB | 3.6 |
| Receiver noise temperature, Kelvins | 374 |
| Weight (antenna and receiver), kg | 15 kg |
| Voltage, vdc | 18 to 32 |
| Power, watts maximum | 100 |
| Dimensions: antenna housing | 60x57x42 cm |
| Internal receiver temperature, Celsius | 35 (+/- 0.03) |
| Environmental: temperature | -50 to +50C |
| **Measurement** | **Specifications** |
| Nominal calibrated brightness temperature accuracy | ≈1.5K |
| Integration cycle | ≈3.9 seconds |
| Warm-up time (typical) | 20 minutes |



**Table 2: The RFI mitigation approach was applied to these synthetic spectra to determine the maximum number of RFI peaks and total bandwidth/proportion of the spectrum that can be contaminated with RFI while still returning a result within 2 K of the mean.**

| number of channels / bandwidth (MHz) | maximum number of peaks | total bandwidth (MHz) of RFI affected channels | proportion of ≈150 MHz spectrum |
|---|---|---|---|
| 1 (≈0.391 MHz) | 20 | 7.81 | 5.2% |
| 3 (≈1.17 MHz) | 17 | 19.92 | 13.2% |
| 5 (≈1.95 MHz) | 9 | 17.58 | 11.7% |
| 10 (≈3.91 MHz) | 4 | 15.63 | 10.4% |

**Table 3: Mean $T_B$ with and without RFI mitigation at the Kernen crop research farm for measurements of unfrozen bare soil at 40**
10 **degrees with and without datalogger RFI emission (25-10-2014 3:00:30 UTC).**

| | No RFI mitigation | RFI mitigation |
|---|---|---|
| $T_B$ (H-pol) **without** datalogger emission | 212.2 | 210.0 |
| $T_B$ (V-pol) **without** datalogger emission | 245.8 | 244.2 |
| $T_B$ (H-pol) **with** datalogger emission | 1761.3 | 210.9 |
| $T_B$ (V-pol) **with** datalogger emission | 955.6 | 246.0 |

15 **Table 4: Mean $T_B$ with and without RFI mitigation from airborne measurements over first year sea ice in Nares Strait, Canadian Arctic (23-04-2011 16:25 UTC).**

| | No RFI mitigation | RFI mitigation |
|---|---|---|
| $T_B$ (H-pol) | 247.3 | 244.3 |
| $T_B$ (V-pol) | 261.3 | 255.8 |