# Peer review of "Radio frequency interference mitigating hyperspectral L-band radiometer"

_Geoscientific Instrumentation, Methods and Data Systems, 2016_

## Referee Comment (RC1) · Anonymous Referee #1 · 6 Sep 2016

General comments

The authors points out that RFI is a problem for spaceborne radiometers operating in the protected band and that the equipment that they have used for this paper can be useful in field campaigns aimed at calibrating/validating such satellite missions. However, the instrument used here (1400-1550 MHz) has a significantly wider passband compared to the spaceborne instruments (1400-1427 MHz). The reason for this difference is not clearly stated. Also, it is not clear if/how this difference affects RFI detection within the protected band.

The authors describe how RFI are detected. Though it is important to specify also how the mitigation is implemented. My guess is that once a sample is considered as RFI-affected it is removed, so that the output is the average of all the RFI-free channels.

[Figure]

But this is not stated in the text.

The calibration is well described.

Some questions remain on the detection algorithm.

It is generally a good paper. The authors put quite a lot of effort in the validation of their results on both simulated and actual measurements. A minor revision is needed before it is suitable for publication.
* * *
Specific comments

Introduction

The name of the mission is "Aquarius/SAC-D", and it was launched jointly by NASA and CONAE. Aquarius is the name of the main instrument.

Soil moisture is the objective for SMOS and SMAP, not for Aquarius. Reword for clarity.

ITU regulations allow spurious emissions in the protected band, as long as they are sufficiently weak.

What do the authors mean by "avoid RFI"?

2.1

It would be clearer if a block diagram were included as well.

3.3

P8 L19-20. This sentence is misleading. The natural thermal emission has a Gaussian distribution. However, when looking at it across frequencies one should expect a uniform distribution.

Also, the dataset in Fig.3 is not a Gaussian distribution. It is the sum of two things: thermal noise (Gaussian) and RFI (not Gaussian). Please review the instances where

a Gaussian distribution is mentioned.

The authors state that the inflection point is representative of the mean if there are no RFI. Is it also representative of the mean for the dataset in Fig.3? If not, please review wording.

All the Tbs above the inflection point are considered as RFI? This seems a very conservative approach. If this was the authors' intention, it is probably worth mentioning it explicitly in the text. Also, if only the measurements below the mean are used, the output will underestimate the natural thermal radiation. Can the authors comment on this?

More commonly, RFI are identified as outliers above a threshold set to "mean +-N*standard deviations", possibly computing the mean iteratively. Did the authors also try to implement this approach and compare the results with the proposed approach?

P8 L31-32. It is not a normal distribution if there are RFI.

How do the authors cope with the cases of "extreme RFI contamination"?

P8 L32-P9 L1. This is not clear to me. If only the protected band is considered and there are no RFI, then this method does not detect RFI? Maybe the authors meant that if only the protected band is considered and there are RFI, then this method does not detect them? In case the latter is correct: I would expect the band 1427-1550 MHz to be even more contaminated than the protected band, since it is allocated to active applications. How does RFI contamination outside the protected band affect the performances on the proposed approach?

After RFI is identified, how is it mitigated? Are the corresponding Tbs removed? Replaced by the mean?

4.1

P11 L11-12. Yes, generally sky calibrations are done pointing out of the galactic plane.

The Moon should also be taken into account.

4.2

Figure 7. Can the authors add a line at Tb-mean = 252 K.

It would be interesting to have also a plot of DeltaTb (Tb-mean – 250 K) against the percent of contaminated bandwidth. It would give a more general idea of the performance of this method.
* * *
Technical corrections

P1 L27. Capitalize moisture

P3 L11. Capitalize CPU

P3 L18-19. Replace "Ten microsecond 1024 (512 horizontal and 512 vertical channel)" with "Ten microsecond 1024 channels (512 horizontal and 512 vertical)"

P4 L28. Replace "out" with "of".

P5 L29. Replace "observed" with "made".

P6 L2. Please remember to update the status of this paper if possible

P6 L2-4. Sentence is not clear, please reword.

P6 L11. Correct "durging"

P7 L22. Remove "1)"

P8 L6. Remove "2)"

P8 L6-7. Reword. Suggestion: the optimization results for the coefficients are used as first guess to run . . .

P8 L11. "science-ready".

P8 L17. Anterrieu 2011 is a software approach, not hardware.

P8 L21-22. Please reword

P9 L14-19. Summarize this paragraph and refer the reader to section 4.2 where all the details should be included (e.g. why below 2 K is OK)
* * *
[Figure]

---

## Referee Comment (RC2) · Anonymous Referee #2 · 3 Oct 2016

Review of paper GI-2016-27 Toose et al. "Radio Frequency interference mitigating hyperspectral L-band radiometer" General The authors describe an L-band radiometer system designed for ground based or airborne experimental studies on microwave remote sensing. The calibration and RFI mitigation method of the system are described. Examples of calibration accuracy and RFI mitigation are demonstrated by means of actual measured data. The limitations of the method regarding broadband RFI mitigation are acknowledged. The paper is well written and generally to the point. The described system is to my knowledge a unique piece of equipment, and many of the features incorporated in its design may be useful when planning for similar future systems (surface based or satellite). In this regard, the authors could add a block diagram of the system to improve the description of the hardware. The study should be of interest to the remote sensing community. I recommend publication in GI after the following

minor comments have been addressed Minor comments 1. P2 lines 5-17. The authors could add some discussion regarding the sensitivity of the different spaceborne sensors (SMOS, SMAP, Aquarius) to RFI. SMOS is the most vulnerable due the applied imaging technique, and also because during design of SMOS, the RFI problem at L-band was not seen as acute. With SMAP, some precautions and mitigation steps could be undertaken (see Bradley et al., 2010; Misra et al., 2013). These could be acknowledged and shortly discussed. 2. Section 2.1, lines 10-20: could you add a block diagram of the radiometer? This would support the text which is now a bit hard to follow. 3. P3, lines 24-26: "Because of filter roll...are utilized" the sentence is a bit too complex and difficult to grasp. Please revise & clarify and maybe split in two. 4. P7 lines 18-19: 5 K is quite a crude approximation for sky TB at L-band. While this is fine for assessing the "sky contribution" via ground reflectance to observations of surface TB, for absolute calibration purposes one would prefer to use a precise value. The authors return to this subject in section 4.1. I would suggest calculating a precise value by means of a model of sky TB at L-band. I it is not possible for this study, the authors would need to provide more justification as to why ∼5K is suitable. One possibility would be to apply a range of realistic values in the calibration and analyze the effects. 5. P7 lines 29-30 "...leading to a difference of 10 K," difference in what? The measured TB (of the ambient target) I guess, but please specify for clarity. 6. Figure 4: please use the same range in y-axes for H and V pol figures. This would highlight larger T_NDTC and Offset peaks at H-pol. Is there any reason for this difference between the polarizations? Perhaps this is discussed somewhere but I missed it. 7. P10 lines 11-15. "Using the results...over course of the campaign." Very long sentence and as a result quite difficult to grasp. Please revise.

Editorial 1. Figure 4: 'T' missing in y-axis label of T_NDTC (H-pol) vs. frequency

---

## Author Comment (AC1) · 4 Nov 2016

The authors would like to thank the reviewers for their time and effort and constructive feedback. We have tried to address all comments and concerns and feel that the manuscript is now much improved. We have uploaded a response to reviewers PDF documents that provides a response to each and every comment from reviewer #1 and #2 as the supplement file associated with this comment. Within the supplement document, the original reviewer comments have been highlighted in blue. Our responses are in black. All changes/additions to the text have been italicized with page and line numbers indicated (referenced to the PDF version of the revised MS Word manuscript with track changes on - uploaded as a second supplement in the author's following comment titled: 'Revised Manuscript - Track Changes').

[Figure]

Please also note the supplement to this comment:
http://www.geosci-instrum-method-data-syst-discuss.net/gi-2016-27/gi-2016-27-AC1-supplement.pdf

**Supplement:**

General comments

The authors points out that RFI is a problem for spaceborne radiometers operating in the protected band and that the equipment that they have used for this paper can be useful in field campaigns aimed at calibrating/validating such satellite missions. However, the instrument used here (1400-1550 MHz) has a significantly wider passband compared to the spaceborne instruments (1400-1427 MHz). The reason for this difference is not clearly stated. Also, it is not clear if/how this difference affects RFI detection within the protected band.

The authors describe how RFI are detected. Though it is important to specify also how the mitigation is implemented. My guess is that once a sample is considered as RFI affected it is removed, so that the output is the average of all the RFI-free channels. But this is not stated in the text.

The calibration is well described.

Some questions remain on the detection algorithm.

It is generally a good paper. The authors put quite a lot of effort in the validation of their results on both simulated and actual measurements. A minor revision is needed before it is suitable for publication.
* * *
Specific comments

Introduction

R1-C1: The name of the mission is "Aquarius/SAC-D", and it was launched jointly by NASA and CONAE. Aquarius is the name of the main instrument.

P1 L28-29 – changed text to – 'The *NASA Aquarius instrument on board the Argentine SAC-D spacecraft*, acquired L-Band observations between September 2012 to July 2015 (Lagerloef et al., 2013)'

R1-C2: Soil moisture is the objective for SMOS and SMAP, not for Aquarius.

P1 L31: – changed text to – 'In addition *to monitoring soil moisture and sea surface salinity*, these missions also provide useful measurements for cryospheric applications…'

R1-C3: Reword for clarity. ITU regulations allow spurious emissions in the protected band, as long as they are sufficiently weak.

P2 L10-14 – changed text to – '…has regulated that the frequency allocation of 1400-1427 MHz be dedicated to passive remote sensing from space and radio astronomy research and that all other emissions within this band be prohibited *or limited to maximum permitted emission power levels (ITU, 2012)*. Illegal sources, in-addition to spurious and harmonic emissions and other unwarranted transmissions, often violate these reserved *bands or exceed the maximum permitted emission power levels,* producing RFI.'

R1-C4: What do the authors mean by "avoid RFI"?

– response to reviewer – One further method of dealing with RFI while trying to conduct L-Band observations using airborne or ground-based instruments is simply to avoid an area known for RFI emissions. The authors acknowledge that the avoidance method is not an option for spaceborne sensors, and as such the term 'avoid' has been removed.

Section 2.1

R1-C5: It would be clearer if a block diagram were included as well.

– response to reviewer – This information is proprietary to Radiometrics Inc. and is not available for publication.

Section 3.3

R1-C6: P8 L19-20. This sentence is misleading. The natural thermal emission has a Gaussian distribution. However, when looking at it across frequencies one should expect a uniform distribution.

P9 Line 25-28 : - changed text to – '*This hyperspectral radiometer system assumes that the natural $T_B$ of a scene is a Gaussian distribution and that the thermal spectra from the observed scene are a near constant $T_B$ across the 1400-1550 MHz bandpass, with minimal observed variability due to the receiver's noise equivalent temperature difference ($\Delta T$), calibration error and some small variance from within the scene (see Table 1.)*

R1-C7: Also, the dataset in Fig.3 is not a Gaussian distribution. It is the sum of two things: thermal noise (Gaussian) and RFI (not Gaussian). Please review the instances where a Gaussian distribution is mentioned.

P10 Line 12-13 : - changed to text – 'Fig. 3 illustrates the horizontal polarized TBs of one integration cycle of the ambient calibration target (251 K) shown in Fig. 2, *which is a sum of the natural thermal signal (Gaussian) and RFI (not Gaussian).*'

R1-C8: The authors state that the inflection point is representative of the mean if there are no RFI. Is it also representative of the mean for the dataset in Fig.3? If not, please review wording.

– response to reviewer – No, the inflection point is not representative of the mean for this dataset because, the mean would be influenced by the RFI-affected channels. We clarify in the text:

P10 Line 14-16: - changed text to – 'The red dot in Fig. 3 is the inflection point *(251.9 K), representative of the mean $T_B$ of the same scene without RFI. The mean value of the full spectrum in Fig. 3 (RFI included), is 255.2 K.'*

R1-C9: All the $T_B$s above the inflection point are considered as RFI? This seems a very conservative approach. If this was the authors' intention, it is probably worth mentioning it explicitly in the text. Also, if only the measurements below the mean are used, the output will underestimate the natural thermal radiation. Can the authors comment on this?

 – response to reviewer – Not all the $T_B$s above the inflection point are identified as RFI, however, there is no way of identifying the exact number of contaminated channels because of the unknown nature/source of the RFI contamination and the potential for subtle RFI intrusions. This exact problem lead the authors to develop our RFI mitigation approach that identifies a single representative $T_B$ value for each measurement as occurring at the point of inflection where the 2nd derivative of the 3rd order polynomial of sort-rank versus $T_B$ goes from negative to positive, which agrees well with the mean $T_B$ of an RFI-free scene of a natural thermal emission. The entire spectrum is used to identify the point of inflection, using the sort-rank versus $T_B$ approach, but the values of each channel are not used in the calculation of the mean $T_B$ of the scene because as the reviewer mentions, the $T_B$s above the inflection point are contaminated by RFI (the exact number of channels and strength of RFI are unknown), and limiting the results to those channels below the inflection point would skew the mean TB too low. Our testing with synthetic data (Section 4.2) suggests that the agreement between the inflection point identified $T_B$ and that of the mean $T_B$ for a clean RFI-scene is maintained (to within 2 K), as long as the number of channels contaminated does not exceed ~9% of the spectrum. As the number of contaminated channels exceeds this percentage, the agreement degrades - See Figure 7.

P9-10 – changed text to – See response to reviewer R1-C14 where we clarify the inflection point approach for RFI mitigation below:

R1-C10: More commonly, RFI are identified as outliers above a threshold set to "mean +- N*standard deviations", possibly computing the mean iteratively. Did the authors also try to implement this approach and compare the results with the proposed approach?

– The authors developed this novel RFI mitigation approach because of its usefulness for separating out the subtle RFI contamination, receiver noise and calibration errors within the spectrum from the expected thermal signal. We think we have successfully demonstrated that our RFI mitigation technique works using both modelled and measured L-Band data. Comparing our results with the 'mean +- N*standard deviations' RFI removal method is beyond the scope of this paper.

R1-C11 : It is not a normal distribution if there are RFI.

P10 L19-21 – change text to – *'The inflection point $T_B$ become less representative of the scene as the spectrum being analyzed becomes exceedingly non-normal…'*

R1-C11 : How do the authors cope with the cases of "extreme RFI contamination"?

P10 L21-23 – change text to – '…*due to over-whelming* RFI contamination, *which results in unrealistic TB values that are relatively easy to detect using a visual inspection of the spectrum and/or of the time series data plots.*'

R1C12: P8 L32-P9 L1. This is not clear to me. If only the protected band is considered and there are no RFI, then this method does not detect RFI? Maybe the authors meant that if only the protected band is considered and there are RFI, then this method does not detect them? –

response to reviewer - If the data to be processed is a uniform short-tailed distribution, then an error during the RFI mitigation processing can occur where the inflection point is identified outside the bounds of the spectrum. If this occurs during the RFI mitigation processing, the midpoint of the spectrum is chosen because it is a highly efficient estimator of the mean, given a small sample of a sufficiently uniform distribution.

P10 Line 25-26 - changed text to – '*In instances where a narrow spectrum is employed and the scene is RFI-free, the midpoint of the spectrum is chosen to represent the mean of the scene, rather than the inflection point.*'

R1-C13 : In case the latter is correct: I would expect the band 1427-1550 MHz to be even more contaminated than the protected band, since it is allocated to active applications. How does RFI contamination outside the protected band affect the performances on the proposed approach? –

response to reviewer – The reviewer is correct, the frequency of observed RFI is higher outside the protected band, however, this higher frequency of observed RFI outside the protected band does not affect the performance of the RFI mitigation approach, as long as the percent of contaminated bandwidth does not become exceedingly large (see section 4.2). However, if a persistent and broadband RFI signal is observed outside the protected band, the RFI mitigation can be applied to a smaller bandwidth (for example: 1400-1475 MHz). The advantage of initially observing over a larger bandwidth is to increase the chance of observing RFI-free channels incase the protected band is contaminated, but as a result, we will record more frequent (legal) RFI occurrences outside the protected band.

P9 Line 21-25 - changed text to – '*The advantage of observing over a larger bandwidth increases the likelihood of observing RFI-free channels, but also results in the recording of more frequent (legal) RFI occurrences outside the protected band. However, this higher frequency of observed RFI outside the protected band does not affect the performance of the RFI mitigation approach described herein, as long as the percent of contaminated bandwidth does not become exceedingly large (see section 4.2).*'

R1-C14 : After RFI is identified, how is it mitigated? Are the corresponding Tbs removed? Replaced by the mean?

– response to reviewer – The authors have tried to re-word/clarify the description of our RFI mitigation technique to better explain our methodology. Our RFI mitigation approach does not remove channels, or calculate a mean $T_B$ based on channels identified as RFI-free. We simply use the $T_B$ value present at the inflection point where the 2nd derivative of the 3rd order polynomial of sort-rank versus $T_B$ goes from negative to positive to represent the scene $T_B$.

P9-10 Line 29-11 – changed text to – 'A simple but effective method of separating out the thermal *signal plus the receiver noise and calibration uncertainty* from RFI-contaminated channels is to sort the

*observed T$_B$ spectra* in ascending order. The thermal channels sort to the low values, *with the spectra gradually increasing by several Kelvin due to the receiver noise, calibration errors and variability within the scene*. The RFI-contaminated channels are then identified where the brightness begins to rise out of the expected thermal spectrum. *However, it is challenging to identify the exact point at which the RFI begins this rise out of the expected thermal spectrum because of the unknown nature/source of the RFI contamination. This dilemma, lead the authors to develop an RFI mitigation approach that identifies a representative RFI-free T$_B$ value for this spectrum*, using a 3rd order polynomial of *the sorted T$_B$ spectra in ascending order.* The 2nd derivative of the slope of this cubic polynomial is derived. For a clean thermal spectrum with random noise, the mean value is a close approximation of the TB at the inflection point where the 2nd derivative goes from negative to positive. *For a spectrum contaminated with RFI, the mean value of the spectrum is skewed higher by the RFI affected channels, however, the T$_B$ at the inflection point where the 2nd derivative of the 3rd order polynomial of sort-rank versus T$_B$ goes from negative to positive is still representative of the mean value of the same scene without RFI (within ≈2 K for a spectrum contamination of up to ≈9%: See section 4.2), and therefore the T$_B$ value at the inflection is used for subsequent analysis and is considered the RFI mitigated result.*'

Section 4.1

R1-C15 : P11 L11-12. Yes, generally sky calibrations are done pointing out of the galactic plane. The Moon should also be taken into account.

P12-13 Line 34-2 – change text to – 'The higher error in the sky measurements could be related to the fact that the sky emission might vary slightly depending upon the observed portion of the sky, *due to the variability in sky background temperatures measured while the antenna beamwidth crosses the galactic plane (1-3 K) and due to the potential contributions from the sun and moon (Delahaye et al., 2002; Le Vine et al., 2005).*'

Delahaye, J.-Y., Golé, P., and Waldteufel, P.: Calibration error of L-band sky-looking ground-based radiometers, Radio Sci., 37(1), 11-1 – 11-11, 2002.

Le Vine, D. M., Abraham, S., Kerr, Y., Wilson, W.J., Skou, N. : Comparison of Model Prediction With Measurements of Galactic Background Noise at L-Band. IEEE T. Geosci. Remote, 43 (9), 2018-2023, 2005

Section 4.2
* * *
Technical corrections
 - DONE
 - DONE

with "Ten microsecond 1024 channels (512 horizontal and 512 vertical)" - DONE
 - DONE

P5 L29. Replace "observed" with "made". - DONE
P6 L2. Please remember to update the status of this paper if possible – Still in review as of October 17[th] 2016
P6 L2-4. Sentence is not clear, please reword. – response to reviewer – The purpose of this sentence was to highlight that the L-Band radiometer system was moved between multiple locations many times, highlighting the unique portability of the system, as well as the ability to successfully mitigate changing RFI sources between sites.

P7 Line 2-7 - changed text to – 'These temporally continuous measurements *recorded in the same location*, were augmented with monthly visits *to multiple sites with varying soil conditions and potentially changing RFI contributions,* across the Kenaston/Brightwater Creek soil monitoring network *85 km south of Saskatoon* ($\approx$51.3° N; $\approx$106.5° W). Between October 23rd, 2014 and April 24th, 2015, 16 three-target calibrations were measured between these two research sites…'

P6 L11. Correct "durging" - DONE
P7 L22. Remove "1)" - DONE
P8 L6. Remove "2)" - DONE
P8 L6-7. Reword. Suggestion: the optimization results for the coefficients are used as first guess to run : : :

P9 Line 8-9 - changed text to – 'Secondly, the optimization results for the coefficients T_ND (0°C) and Offset(0°C) from the first post-processing step, are used as a first guess to run the inversion model again, …'

P8 L11. "science-ready". - DONE
P8 L17. Anterrieu 2011 is a software approach, not hardware. – Removed reference
P8 L21-22. Please reword

P9-10 Line 31-5 – changed text to – 'The thermal channels sort to the low values, *with the spectra gradually increasing by several Kelvin due to the receiver noise, calibration errors and variability within the scene.* The RFI-contaminated channels are then identified where the brightness begins to rise out of the expected thermal spectrum. *However, it is challenging to identify the exact point at which the RFI begins this rise out of the expected thermal spectrum because of the unknown nature/source of the RFI contamination. This dilemma, lead the authors to develop an RFI mitigation approach that identifies a representative RFI-free TB value for this spectrum,* using a 3rd order polynomial of *the sorted TB spectra in ascending order.*'

P9 L14-19. Summarize this paragraph and refer the reader to section 4.2 where all the details should be included (e.g. why below 2 K is OK) – response to reviewer – The authors feel that the methods section is the more appropriate section to describe the assessment criteria. The authors have added the following reference to the results section 4.2 to re-direct the reader to section 3.4 for more details on how/why the 2 K assessment criteria was chosen.

P13 Line 10-11 - change to text – 'If the RFI mitigation results were within 2 K, of the mean 250 K of the synthetic spectrum, then the RFI mitigation approach was considered successful *(See section 3.4 for successful assessment criteria).*'

Review of paper GI-2016-27 Toose et al. "Radio Frequency interference mitigating hyperspectral L-band radiometer"

General

The authors describe an L-band radiometer system designed for ground based or airborne experimental studies on microwave remote sensing. The calibration and RFI mitigation method of the system are described. Examples of calibration accuracy and RFI mitigation are demonstrated by means of actual measured data. The limitations of the method regarding broadband RFI mitigation are acknowledged. The paper is well written and generally to the point. The described system is to my knowledge a unique piece of equipment, and many of the features incorporated in its design may be useful when planning for similar future systems (surface based or satellite). In this regard, the authors could add a block diagram of the system to improve the description of the hardware. The study should be of interest to the remote sensing community. I recommend publication in GI after the following minor comments have been addressed:

Minor comments

R2-C1: P2 lines 5-17. The authors could add some discussion regarding the sensitivity of the different spaceborne sensors (SMOS, SMAP, Aquarius) to RFI. SMOS is the most vulnerable due the applied imaging technique, and also because during design of SMOS, the RFI problem at L-band was not seen as acute. With SMAP, some precautions and mitigation steps could be undertaken (see Bradley et al., 2010; Misra et al., 2013). These could be acknowledged and shortly discussed.

– response to reviewer – The authors have added a brief description of the RFI detection/mitigation strategies employed by these three spaceborne L-band radiometer instrument/missions to our introduction section.

P2-3 Line 16-5 - changed text to - '*A number of different hardware and processing approaches have been explored and implemented for handling RFI intrusions for a variety of L-Band radiometer systems* (Guner et al., 2007; Forte et al., 2011; Pardé et al., 2011; Misra et al., 2009). *The SMOS mission was the first L-Band spaceborne mission launched in several decades, and thus was the least prepared for dealing with the prevalence of RFI contamination. Anterrieu and Khazâal (2011), and Khazâal et al., (2014), developed post-launch methods to flag SMOS snapshots as RFI-contaminated, by identifying outliers in temporally averaged Level L1a data, and by evaluating the kurtosis for each snapshot. These methods simply identify RFI contaminated data with no mitigation applied, and were developed specifically for SMOS due to the unique SMOS hardware and software architecture, which does not archive the radiometric signals at their highest temporal resolution, limiting standard approaches for detecting RFI contamination (Anterrieu and Khazâal, 2011). The prevalence of L-Band RFI was well-acknowledged prior to the launch of the Aquarius instrument, and thus the radiometer was designed with RFI mitigating capabilities. Le Vine et al., (2014), describe the rapid sampling 'glitch detection' algorithm employed by*

*the Aquarius L-Band instrument that differs from SMOS by rapidly sampling the scene many times in the time required for the antenna to move half the width of an image pixel, so that potentially those samples corrupted with RFI could be identified by comparing these measurements to preset thresholds. If the samples exceed the thresholds, they are flagged as RFI and are not included in computing the mean of all measured samples for an image pixel. The recently launched SMAP mission employs a more advanced RFI mitigation technology that was not available to previous generation spaceborne L-Band radiometers, involving both space-flight instrument hardware and ground-based processing algorithms. Piepmeier et al., (2014), describe the hardware and processing algorithms of the SMAP radiometer that outputs the first four raw moments of the receiver-system noise voltage in 16 frequency channels for measuring noise temperature and kurtosis, as well as cross-correlation products for measuring the third and fourth Stokes parameters. The ground-based processing algorithms utilize several detectors to identify RFI in the frequency, time, statistical, and polarization domains measured by the instrument. These detectors are then utilized to remove the contaminated time / bandwidth portions of the observation. Therefore, the Aquarius instrument and SMAP radiometer have the ability to flag RFI affected samples, as well as mitigate the influence of RFI, and provide a measure of the level of RFI contamination.'*

R2-C2:  Section 2.1, lines 10-20: could you add a block diagram of the radiometer? This would support the text which is now a bit hard to follow.

– response to reviewer –  This information is proprietary to Radiometrics Inc. and is not available for publication

R2-C3:  P3, lines 24-26: "Because of filter roll: : :are utilized" the sentence is a bit too complex and difficult to grasp. Please revise & clarify and maybe split in two.

P4 Line 24-27 – changed text to – *'To reduce the effects of spectral leakage from adjacent bands of strong RFI, the channels near the edges of the* 200 MHz receiver bandwidth are not used for analysis, and instead only 385 channels and ≈150 MHz bandwidth (1400 to 1550.5 MHz) are utilized.'

R2-C4:  P7 lines 18-19: 5 K is quite a crude approximation for sky TB at L-band. While this is fine for assessing the "sky contribution" via ground reflectance to observations of surface TB, for absolute calibration purposes one would prefer to use a precise value. The authors return to this subject in section 4.1. I would suggest calculating a precise value by means of a model of sky TB at L-band. If it is not possible for this study, the authors would need to provide more justification as to why 5K is suitable. One possibility would be to apply a range of realistic values in the calibration and analyze the effects.

– response to reviewer – A range of observed and modeled sky background temperatures have been reported in the literature. These publications agree on the 2.7 K cosmic background radiation, and differ slightly on the contribution of the atmosphere at L-Band (1.5-1.9 K) and 1-3 K for the contribution of the galatic background (smoothed by large beamwidth antennas).  The authors believe that the following references provide justification on the use of the 5 K temperature of the sky for use in the radiometer calibration procedure:

1. Pellarin et al. (2016) reported the average background sky temperatures of 4.44 and 4.46 K (H, V) with standard deviations of 0.27 and 0.29 K for three years of sky observations at 1200 UTC at 135° relative to nadir in a westerly direction near Grenoble, France.

2. Delahaye et al. (2002) calculated the best calibration orientation for a sky-looking radiometer at medium northern latitude would be 0° in azimuth (northward) and an elevation equal to the radiometer's latitude. They found that the computed total sky noise contribution would be 6.6 K, with 24 hour variations of ±0.2 K and a maximum bias of ±0.6 K. Their results are valid for the whole year, assuming low to moderate solar activity and no rain.

3. Le Vine et al., (2005) compared modeled and measured background sky temperatures from multiple L-Band radiometers. The observed sky background temperatures were typically between 5 and 6 K, and varied an additional 1-3 K when crossing the galatic plane.

4. Lemmetyinen et al., (2016) report that using ≈ 5 K as the reference sky temperature, the ELBARA-II showed a standard deviation of less than 0.4 K at both polarizations while measuring the sky at zenith for the entire 6 year campaign.

P8 Line 18-20 – changed text to – 'The reference sky TB at L-Band was considered to be ≈5 K for *both* polarizations *based on previously published data from both measured and modelled sources* (Pellarin et al., 2016; Lemmetyinen et al., 2016; *Le Vine et al., 2005; Delahaye et al., 2002*).'

R2-C5: . P7 lines 29-30 ": : :leading to a difference of 10 K," difference in what? The measured TB (of the ambient target) I guess, but please specify for clarity.

P8 Line 30-32 – changed text to – 'The minimum and maximum observed $T_{case}$ values were used as inputs (-18.1 and 23.5 °C), leading to a difference of ≈10 K (269.4 K versus 259.3 K) *in the calculated $T_B$ of the ambient calibration target*, highlighting the importance of applying a temperature *correction to reflect changes in the environmental operating temperatures of the radiometers*.'

R2-C6:  Figure 4: please use the same range in y-axes for H and V pol figures. This would highlight larger T_NDTC and Offset peaks at H-pol. – DONE

Is there any reason for this difference between the polarizations? Perhaps this is discussed somewhere but I missed it.

P11 Line 27-28 – changed text to – *'RFI intrusions can be polarization specific, as illustrated by the differences in the dashed-green line between the H-pol and V-pol plots.'*

R2-C7 : 7. P10 lines 11-15. "Using the results: : :over course of the campaign." Very long sentence and as a result quite difficult to grasp. Please revise.

P12 Line 2-5 – changed text to – '… the smoothed results for the coefficients T_ND (0°C) and Offset(0°C) are used as a first guess, to run the inversion model again. *A set of new calibration coefficients for T_ND (0°C) and Offset (0°C), for each date of the 16 three-target calibrations, are produced. These coefficients are used* to correct for sensor drift of the radiometer over course of the campaign (Fig. 5: red dots - for the 1451 MHz channel).'

R2-C8: Editorial 1. Figure 4: 'T' missing in y-axis label of T_NDTC (H-pol) vs. frequency - DONE

---

## Author Comment (AC2) · 4 Nov 2016

The authors have made changes to the original manuscript based on the comments/feedback from the reviewers, in-addition to some minor editorial changes. All changes were made in MS Word with the track-changes-on, highlighting all additions/deletions. A PDF version of this revised manuscript with track-changes-on has been uploaded as a supplement to this comment. The previous comment posted by the authors also had an uploaded supplement that referenced the revised manuscript.

Please also note the supplement to this comment:

http://www.geosci-instrum-method-data-syst-discuss.net/gi-2016-27/gi-2016-27-AC2-supplement.pdf

**Supplement:**

**Radio frequency interference mitigating hyperspectral L-band radiometer**

Peter Toose[1], Alexandre Roy[2], Frederick Solheim[3], Chris Derksen[1], Tom Watts[4], Alain Royer[2] and Anne Walker[1]

[1] Climate Research Division, Environment and Climate Change Canada, Toronto, Ontario, M3H 5T4 Canada.
[2] Centre d'Applications et de Recherches en Télédétection, Université de Sherbrooke, Sherbrooke, Québec, J1K 2R1, Canada
[3] Dakota Ridge Research and Development, Boulder, Colorado, 80303, United States.
[4] Department of Geography, Northumbria University, Newcastle upon Tyne, NE1 8ST, United Kingdom.

*Correspondence to*: Peter Toose (Peter.Toose@canada.ca)

**Abstract.** Radio Frequency Interference (RFI) can significantly contaminate the measured radiometric signal of current spaceborne L-band passive microwave radiometers. These spaceborne radiometers operate within the protected passive remote sensing and radio astronomy frequency allocation of 1400-1427 MHz, but despite this are still subjected to frequent RFI intrusions. We present a unique surface-based/airborne hyperspectral 385 channel, dual polarization, L-band Fourier transform, RFI detecting radiometer designed with a frequency range from 1400 through ≈1550 MHz. The extended frequency range was intended to increase the likelihood of detecting adjacent RFI-free channels to increase the signal, and therefore increase the thermal resolution, of the radiometer instrument. The external instrument calibration uses three targets (sky, ambient, and warm) and validation from independent stability measurements shows a mean absolute error (MAE) of 1.0 K for ambient and warm targets, while the MAE is 1.5 K for sky. A simple but effective RFI removal method which exploits the large number of frequency channels is also described. This method separates the desired thermal emission from RFI intrusions, and was evaluated with synthetic microwave spectra generated using a Monte Carlo approach and validated with surface-based and airborne experimental measurements.

**Keywords:** surface-based / airborne radiometer, L-Band, radio frequency interference, RFI mitigation

**1 Introduction**

A number of spaceborne L-Band passive microwave radiometer missions were successfully launched in recent years for global monitoring of soil moisture and sea surface salinity. The European Space Agency Soil Moisture and Ocean Salinity (SMOS) mission (Kerr et al., 2010), was launched in November 2009 and continues to operate. The  NASA Aquarius  instrument  on board the Argentine SAC-D spacecraft, acquired L-Band observations between September 2012 to July 2015 (Lagerloef et al., 2013), and the NASA Soil Moisture Active Passive (SMAP) satellite was launched in January 2015 (Entekhabi et al., 2015). In addition to monitoring soil moisture and sea surface salinity, these

missions also provide useful measurements for cryospheric applications including monitoring the freeze/thaw state of the land surface (Rautiainen et al., 2016; Rautiainen et al., 2014; Roy et al., 2015), estimating snow density and ground permittivity (Schwank et al., 2015; Lemmetyinen et al., 2016), and retrieving the thickness of thin sea ice (Kaleschke et al., 2016; Kaleschke et al., 2012).

Even though the SMOS, Aquarius, and SMAP radiometer bandwidths fall within a protected band (1400-1427 MHz), significant levels of radio frequency interference (RFI) caused by anthropogenic sources of radiation are commonly observed in satellite L-Band measurements (Oliva et al., 2016; Le Vine et al. 2014; Piepmeier et al., 2014; Askoy and Johnson, 2013). The United Nations provision 5.340 of Radio Regulations of the International Telecommunication Union -
10  Radiocommunications Sector (ITU-R), has regulated that the frequency allocation of 1400-1427 MHz be dedicated to passive remote sensing from space and radio astronomy research and that all other emissions within this band be prohibited or limited to maximum permitted emission power levels (ITU, 2012). Illegal sources, in-addition to spurious and harmonic emissions and other unwarranted transmissions, often violate these reserved bands or exceed the maximum permitted emission power levels, producing RFI. The natural thermal emissions in these protected wavebands are orders of magnitude
15  lower in power than active RFI sources; therefore, such RFI intrusions can contaminate and even blind the passive observations. While aA number of different hardware and processing approaches have been explored and implemented for handling RFI intrusions for a variety of L-Band radiometer systems (Guner et al., 2007; Anterrieu, 2011; Forte et al., 2011; Pardé et al., 2011; Misra et al., 2009). The SMOS mission was the first L-Band spaceborne mission launched in several decades, and thus was the least prepared for dealing with the prevalence of RFI contamination. Anterrieu and Khazâal
20  (2011), and Khazâal et al., (2014), developed post-launch methods to flag SMOS snapshots as RFI-contaminated, by identifying outliers in temporally averaged Level L1a data, and by evaluating the kurtosis for each snapshot. These methods simply identify RFI contaminated data with no mitigation applied, and were developed specifically for SMOS due to the unique SMOS hardware and software architecture, which does not archive the radiometric signals at their highest temporal resolution, limiting standard approaches for detecting RFI contamination (Anterrieu and Khazâal, 2011). The prevalence of
25  L-Band RFI was well-acknowledged prior to the launch of the Aquarius instrument, and thus the radiometer was designed with RFI mitigating capabilities. Le Vine et al., (2014), describe the rapid sampling 'glitch detection' algorithm employed by the Aquarius L-Band instrument that differs from SMOS by rapidly sampling the scene many times in the time required for the antenna to move half the width of an image pixel, so that potentially those samples corrupted with RFI could be identified by comparing these measurements to preset thresholds. If the samples exceed the thresholds, they are flagged as
30  RFI and are not included in computing the mean of all measured samples for an image pixel. The recently launched SMAP mission employs a more advanced RFI mitigation technology that was not available to previous generation spaceborne L-Band radiometers, involving both space-flight instrument hardware and ground-based processing algorithms. Piepmeier et al., (2014), describe the hardware and processing algorithms of the SMAP radiometer that outputs the first four raw moments of the receiver-system noise voltage in 16 frequency channels for measuring noise temperature and kurtosis, as well as crosscorrelation products for measuring the third and fourth Stokes parameters. The ground-based processing algorithms utilize several detectors to identify RFI in the frequency, time, statistical, and polarization domains measured by the instrument. These detectors are then utilized to remove the contaminated time / bandwidth portions of the observation. Therefore, the Aquarius instrument and SMAP radiometer have the ability to flag RFI affected samples, as well as mitigate the influence of RFI, and provide a measure of the level of RFI contamination.

[revised manuscript text omitted]

**3.3 Hyperspectral RFI mitigation approach**

A number of L-Band RFI identification and mitigation methods have previously been proposed and developed into hardware for both airborne and satellite systems (Guner et al., 2007;  Forte et al., 2011; Pardé et al., 2011; Misra et al., 2009), but the hyperspectral system described in this paper has the capability of identifying RFI-free channels over an expanded 1400-1550 MHz spectrum. The advantage of observing over a larger bandwidth increases the likelihood of observing RFI-free channels, but also results in the recording of more frequent (legal) RFI occurrences outside the protected band. However, this higher frequency of observed RFI outside the protected band does not affect the performance of the RFI mitigation approach described herein, as long as the percent of contaminated bandwidth does not become exceedingly large (see section 4.2). This hyperspectral radiometer system assumes that the natural $T_B$ of a scene is a Gaussian distribution and that the thermal spectra from the observed scene are a near constant $T_B$ across the 1400-1550 MHz bandpass, with minimal observed variability due to the receiver's noise equivalent temperature difference ($\Delta T$), calibration error and some small variance from within the scene (see Table 1).  A simple but effective method of separating out the thermal  signal plus the receiver noise and calibration uncertainty from RFI-contaminated channels is to sort the observed $T_B$s  in ascending order. The thermal channels sort to the low values, with the spectra gradually increasing by several Kelvin due to the receiver noise, calibration errors and variability within the scene.  The RFI-contaminated channels are then identified

where the brightness begins to rise out of the expected thermal spectrum. However, it is challenging to identify the exact point at which the RFI begins this rise out of the expected thermal spectrum because of the unknown nature/source of the RFI contamination. This dilemma, lead the authors to develop an RFI mitigation approach that identifies  a representative RFI-free $T_B$ value  for this spectrum, using a $3^{rd}$ order polynomial of the sorted $T_B$ spectra in ascending order. The $2^{nd}$ derivative of the slope of this cubic polynomial is derived. For a clean thermal spectrum with random noise, the mean value is a close approximation of the $T_B$ at the inflection point where the $2^{nd}$ derivative goes from negative to positive. For a spectrum contaminated with RFI, the mean value of the spectrum is skewed higher by the RFI affected channels, however, the $T_B$ at the inflection point where the $2^{nd}$ derivative of the $3^{rd}$ order polynomial of sort-rank versus $T_B$ goes from negative to positive is still representative of the mean value of the same scene without RFI (within ≈2 K for a spectrum contamination of up to ≈9%: See section 4.2), and therefore the $T_B$ value at the inflection is used for subsequent analysis and is considered the RFI mitigated result.  Fig. 3 illustrates the horizontal polarized $T_B$s of one integration cycle of the ambient calibration target (251 K) shown in Fig. 2, which is a sum of the natural thermal signal (Gaussian) and RFI (not Gaussian). The data are sorted into ascending order represented by the blue-dashed line. The red dot in Fig. 3 is the inflection point (251.9 K), representative of the mean $T_B$ of the same scene without RFI. The mean value of the full spectrum in Fig. 3 (RFI included), is 255.2 K. To the right of the inflection point are the natural thermal spectrum values larger than the mean $T_B$ value (the high side of the Gaussian distribution of thermal energies), and the RFI affected channels. To the left are the values of the natural thermal spectrum smaller than the mean value, but only the $T_B$ value at the inflection point is used to define the observed $T_B$ of the scene. The inflection point $T_B$ become less representative of the scene as  the spectrum being analyzed  becomes exceedingly non-normal  due to  over-whelming RFI contamination, which results in unrealistic $T_B$ values that are relatively easy to detect using a visual inspection of the spectrum and/or of the time series data plots. The spectrum can also become non-normal from limiting the application of the RFI mitigation approach to a narrow bandwidth , with too few channels creating a uniform, short-tailed distribution when the scene is RFI-free. In instances where a narrow bandwidth is employed and the scene is RFI-free, the midpoint of the spectrum is chosen to represent the mean of the scene, rather than the inflection point. This RFI mitigation approach is applied to each polarization separately, typically for all 385 channels in the 1400-1550 MHz spectrum. All subsequent temporal averaging of the $T_B$ measurements is then applied to the RFI mitigated data.

**3.4 Sensitivity analysis of the RFI mitigation approach using modelled data**

[revised manuscript text omitted]

**Comment [PT1]:** Previous simulatio
was run on entire 200 MHz spectrum, ra
than the 150 MHz spectrum. Updated fig
and addressed reviewer #1's comments.

**Figure 7: For all 1000 replicates of the different combinations of peaks (0 up to 20) and bandwidths (1-3-5-10), a mean $T_B$ and associated standard deviation (error bars) from the RFI mitigation results was calculated and are plotted (solid lines) for comparison to the mean $T_B$ of the synthetic spectra (set to 250 K). The error bars show the standard deviation of the 1000 replicates for each scenario.A horizontal dashed line has been added to both plots highlighting the RFI mitigation results within 2 K of the synthetic spectra mean.**

[Figure]

5 **Figure 8: V-pol (left) and H-pol (right) $T_B$ measured at Kernen crop research farm during datalogger RFI emission (25-10-2014 3:00:00 UTC).**

[Figure]

**Figure 9. V-pol (left) and H-pol (right) $T_B$ measured at Kernen crop research farm without datalogger RFI emission (25-10-2014 3:00:30 UTC). The red line is representative of the RFI-mitigated $T_B$ results.**

[Figure]

**Figure 10. V-pol (left) and H-pol (right) T$_B$ measured over first year sea ice in Nares Strait, Canadian Arctic (23-04-2011 16:25 UTC). Note the V-pol broadband RFI intrusion. The red line is representative of the RFI-mitigated T$_B$ results.**

**Tables**

Table 1: ECCC L-Band radiometer component and measurement specifications

| Component | Specification |
|---|---|
| Radiometer receiver architecture | high sideband down conversion |
| Frequency range, GHz | 1.40 to 1.55 |
| Edge-to-edge IF bandpass, MHz | 150 MHz (hyperspectral mode) or 25 MHz (broadband mode); user selectable |
| Hyperspectral mode | 385 channels of ≈391 KHz width |
| Radiometric resolution, $\Delta T$, Kelvins | |
| single observation, single channel | 2.52 |
| single observation, 385 channels | 0.13 |
| Antenna HPBW, degrees | 30 |
| Sidelobes, -dB | -20 |
| Antenna type | conformal muffin tin antenna |
| Noise figure, dB | 3.6 |
| Receiver noise temperature, Kelvins | 374 |
| Weight (antenna and receiver), kg | 15 kg |
| Voltage, vdc | 18 to 32 |
| Power, watts maximum | 100 |
| Dimensions: antenna housing | 60x57x42 cm |
| Internal receiver temperature, Celsius | 35 (+/- 0.03) |
| Environmental: temperature | -50 to +50C |
| **Measurement** | **Specifications** |
| Nominal calibrated brightness temperature accuracy | ≈1.5K |
| Integration cycle | ≈3.9 seconds |
| Warm-up time (typical) | 20 minutes |

Comment [PT2]: Updated table with new values. Original simulation results were produced from analysing the entire 200 MHz spectrum, instead of the 150 MHz spectrum.

**Table 2**: **The RFI mitigation approach was applied to these synthetic spectra to determine the maximum number of RFI peaks and total bandwidth/proportion of the spectrum that can be contaminated with RFI while still returning a result within 2 K of the mean.**

| number of channels / bandwidth (MHz) | maximum number of peaks | total bandwidth (MHz) of RFI affected channels | proportion of ≈150 MHz spectrum |
|---|---|---|---|
| 1 (≈0.391 MHz) | 20 | 7.8 | 5.2% |
| 3 (≈1.17 MHz) | 11 | 12.9 | 8.6% |
| 5 (≈1.95 MHz) | 6 | 11.7 | 7.8% |
| 10 (≈3.91 MHz) | 3 | 11.7 | 7.8% |

Table 3: Mean $T_B$ with and without RFI mitigation at the Kernen crop research farm for measurements of unfrozen bare soil at 40 degrees with and without datalogger RFI emission (25-10-2014 3:00:30 UTC).

| | No RFI mitigation | RFI mitigation |
|---|---|---|
| $T_B$ (H-pol) **without** datalogger emission | 212.2 | 210.0 |
| $T_B$ (V-pol) **without** datalogger emission | 245.8 | 244.2 |
| $T_B$ (H-pol) **with** datalogger emission | 1761.3 | 210.9 |
| $T_B$ (V-pol) **with** datalogger emission | 955.6 | 246.0 |

Table 4: Mean $T_B$ with and without RFI mitigation from airborne measurements over first year sea ice in Nares Strait, Canadian Arctic (23-04-2011 16:25 UTC).

| | No RFI mitigation | RFI mitigation |
|---|---|---|
| $T_B$ (H-pol) | 247.3 | 244.3 |
| $T_B$ (V-pol) | 261.3 | 255.8 |